# Surfactant control of gas transfer velocity along an offshore coastal transect: results from a laboratory gas exchange tank

R. Pereira[1,2], K. Schneider-Zapp[2,3], R. C. Upstill-Goddard[2]

[1]The Lyell Centre, Heriot-Watt University, Edinburgh, EH14 4AP, UK

[2]School of Marine Science and Technology, Newcastle University, Newcastle upon Tyne, NE1 7RU, UK

[3]Pix4D, EPFL Innovation Park, 1015 Lausanne, Switzerland

*Correspondence to:* R. Pereira (R.Pereira@hw.ac.uk)

**Abstract**

Understanding the physical and biogeochemical controls of air–sea gas exchange is necessary for establishing
biogeochemical models for predicting regional- and global-scale trace gas fluxes and feedbacks.  To this end we report the results of experiments designed to constrain the effect of surfactants in the sea surface microlayer (SML) on the gas transfer velocity ($k_w$; cm hr$^{-1}$), seasonally (2012-2013) along a 20 km coastal transect (North East UK). We measured total surfactant activity (SA), colorimetric dissolved organic matter (CDOM) and chlorophyll-a (Chl-a) in the SML and in sub-surface water (SSW) and we evaluated corresponding $k_w$ values using a custom-designed air-sea gas exchange tank. Temporal SA
variability exceeded its spatial variability. Overall, SA varied five-fold between all samples (0.08 - 0.38 mg L$^{-1}$ T-X-100), being highest in the SML during summer. SML SA enrichment factors (EF's) relative to SSW were ~ 1.0 - 1.9, except for two values (0.75; 0.89: February 2013). The range in corresponding $k_{660}$ ($k_w$ for $CO_2$ in seawater at 20 $^o$C) was 6.8 - 22.0 cm hr$^{-1}$. The film factor $R_{660}$ (the ratio of $k_{660}$ for seawater to $k_{660}$ for "clean", i.e. surfactant-free, laboratory water) was strongly correlated with SML SA (r $\geq$0.70, p $\leq$0.002, each n = 16). High SML SA typically corresponded to $k_{660}$ suppressions ~14 -
51 % relative to clean laboratory water, highlighting strong spatio-temporal gradients in gas exchange due to varying surfactant in these coastal waters. Such variability should be taken account of when evaluating marine trace gas sources and sinks. Total CDOM absorbance (250-450 nm), the CDOM spectral slope ratio ($S_R = S_{275-295} / S_{350-400}$), the 250:365 nm CDOM absorption ratio ($E_2 : E_3$) and Chl-a all indicated spatial and temporal signals in the quantity and composition of organic matter in the SML and SSW. This prompts us to hypothesize that spatio-temporal variation in $R_{660}$ and its
relationship with SA is a consequence of compositional differences in the surfactant fraction of the SML DOM pool that warrants further investigation.

## Introduction

The global budgets of important climate active gases such as carbon dioxide ($CO_2$), nitrous oxide ($N_2O$) and methane ($CH_4$)
have important marine components that are predicted to change in a future climate (Bakker et al., 2014). For $CO_2$, uncertainty over the spatial and temporal variability of its water-side gas transfer velocity ($k_w$) is the greatest obstacle to accurately evaluating its net global rate of air-sea exchange (Takahashi et al., 2009; 2012). Environmental control of $k_w$ is exerted through the modification of turbulent diffusion at the air-sea interface (Upstill-Goddard, 2006).  Wind speed is the most fundamental control but its use for predicting $k_w$ is compromised by high data scatter that is unrelated to
methodological issues (Asher, 2009). The result is several parameterizations (e.g. Nightingale et al., 2000; Wanninkhof, 1992; Wanninkhof et al., 1997; Wanninkhof and McGillis, 1999) with differences that imply variable influences from other factors such as atmospheric stability, sea state, breaking waves, white caps, bubble transport, rain and the presence of surfactants and other organics (Upstill-Goddard, 2006).

The surface ocean boundary with the atmosphere is characterised by the sea surface microlayer (SML) which is $\sim 400$ μm or less deep and is physically and biogeochemically distinct from the underlying water (Cunliffe et al., 2013). Dissolved components and buoyant particles from the underlying water become enriched in the SML by bubble scavenging (Cunliffe at al., 2009; Cunliffe et al., 2013; Gaŝparović et al., 1998; Petrović et al., 2002; Wurl et al., 2011 and Źutić et al., 1981), leading to accelerated rates of microbiological and photochemical processes (Cunliffe et al., 2013; Vodacek et al., 1997; Häder et al., 2011). Material accumulating in the SML includes a range of surface active substances (surfactants) such as transparent exopolymer particles (TEP; Wurl and Holmes, 2008), polysaccharides (Sieburth et al., 1976), lipid-like material (Gaŝparović et al 1998; Kattner and Brockmann 1978; Lass and Friedrichs, 2011), amino acids (Kuznetsova et al., 2004) and chromophoric dissolved organic matter (CDOM; Tilstone et al., 2010). The tendency is for many of these components to be of lower molecular weight than their analogues in the underlying water (Lechtenfeld et al., 2013) and this may be coupled to in-situ primary production (Chin et al., 1998; Passow 2002), allochthonous inputs of terrestrial material of either natural (e.g. Frew et al., 2006) or anthropogenic (Guitart et al., 2007) origin, and the photochemical and/or microbial reworking of higher molecular weight material (Tilstone et al., 2010; Schulz et al., 2013).

The SML is itself overlain by the surface nanolayer (SNL); this is $\sim 1$–10 nm thick and can be a monolayer, it also comprises of surface-active substances and it may be enriched in carbohydrates during summer (Lass et al., 2013). Its physico-chemical properties differ from those of the SML, providing an additional diffusion barrier and modifying the viscoelasticity of the air-sea interface (McKenna and Bock, 2006). This reduces the rate of air-sea gas exchange by wave damping and by attenuating turbulent energy transfer (Liss and Duce, 1997). It is these effects that are manifested in reductions in the value of $k_w$ (McKenna and McGillis, 2004; Salter et al., 2011).

While $k_w$ suppression by surfactants ranging from 5-55% has been observed in the laboratory (Bock et al., 1999; Goldman et al., 1988) and at sea (Brockmann et al., 1982; Salter et al., 2011), this has mostly involved artificial compounds. The role of natural surfactants has remained inadequately quantified due to the complexity of measuring $k_w$ in-situ and the spatial and temporal variability in natural surfactant concentration and composition (Salter et al., 2011). Although Schmidt and Schneider (2011) examined the seasonal variability in $k_w$ associated with changes in surfactant from measurements of $O_2$ transfer and changes in surface tension, there remains a critical knowledge gap with respect to the spatio-temporal variability of $k_w$ linked to surfactant. We therefore estimated the temporal variability in $k_w$ along an offshore gradient in natural surfactant, using a laboratory gas exchange tank custom built for this purpose.

The overarching goal of this study was to derive a fundamental understanding of the spatial and temporal control of $k_w$ variability by surfactant. Our testable hypothesis is that due to surfactant suppression of $k_w$, inverse correlations between $k_w$ and surfactant activity (SA) in the SML should temporally persist in regions where SA shows high spatial variability. A secondary aim was to ascertain whether surfactant accumulation in the SML is strongly linked to primary productivity (using chlorophyll-a as a proxy) and whether CDOM could be used as a quantitative index of SA and $k_w$, given its widespread use in remote sensing platforms.

**Materials and Methods**

Sampling was at five, 5 km spaced stations along the Dove Time Series (DTS; Frid et al., 1999) transect off the North East UK coast, using RV *Princess Royal*. The purpose of the DTS is to monitor long-term changes in the plankton community structure of the North Sea (Clark 2000). For this study we opportunistically sampled at established DTS sampling stations (Fig. 1). At each station the SML and underlying sub-surface water (SSW) were sampled in triplicate. To minimise contamination from RV *Princess Royal* all samples were collected from near the bow whilst stationary and with the bow

positioned upwind. Visual inspection for potential fouling from the research vessel prior to sampling aimed to ensure collection of a representative sample. Following a well-established protocol (Cunliffe et al., 2014), the SML was sampled using a Garrett Screen (Garrett, 1965; mesh 16, wire diameter 0.36 mm, opening 1.25 mm) with an effective surface area of 2025 cm$^2$. At each station we made three Garrett Screen dips, with a typical total yield for all three dips of 400-500 mL, corresponding to an estimated sampling depth of 65-80 μm. The samples were transferred to three 50 mL high-density polyethylene (HDPE) bottles and stored in an on-board refrigerator at 4 ºC (Cunliffe et al., 2013; Schneider-Zapp et al., 2013). SSW (~100 litres) was sampled from a non-toxic supply pump located ~1 m below the water surface and collected in 5 x 20 L HDPE carboys that were pre-cleaned with 20% HCl to remove leachable organics. SSW temperature and salinity were measured *in-situ* (Hanna Instruments, UK). In the absence of directly measured wind speed, monthly mean wind speeds of daily means at 10 m ($U_{10}$) from January 2012 to December 2013 were obtained from the ERA Interim reanalysis dataset supplied by the European Centre for Medium-Range Weather Forecasts (http://www.ecmwf.int/). During our study period $U_{10}$ ranged from 4.7 to 11.4 m s$^{-1}$ with the highest values between October and February and lowest values in June and July. On return to the laboratory, typically within 6 hours of sampling, all samples were stored in a 4 ºC dark cold room according to the procedure of Schneider-Zapp et al., (2013). The SML and SSW were analysed for surfactant activity (SA) and chromophoric dissolved organic matter (CDOM) absorbance and the SSW was additionally analysed for Chlorophyll-a (Chl-a). All analyses were completed within 12 hours.

Surfactant activity (SA) was measured in triplicate by phase-sensitive AC voltammetry (797 VA Computrace: Metrohm, Switzerland) using a hanging mercury drop (Ćosović and Vojvodić, 1982) with sample salinities pre-adjusted to 35.0 via the addition of surfactant-free 3 mol L$^{-1}$ NaCl solution. Calibration was against the non-ionic soluble surfactant Triton T-X-100. All equipment was acid-washed (10% HCl) and analytical grade water-rinsed (18.2 Ohm Milli-Q, Millipore System Inc., USA) prior to use.

Chl-a was determined fluorometrically (Welschmeyer, 1994). In brief, 0.5 L of each seawater sample was filtered through a 0.2 μm, 47 mm diameter Nylon membrane filter (Knefelkamp et al., 2007). The filters were subsequently extracted in acetone and the extract analysed using a Turner Designs Trilogy fluorometer (USA) calibrated with a known pure Chl-a standard (Anacystis nidulans, Sigma Aldrich).

CDOM absorbance (wavelength range 800 - 250 nm) was determined on unfiltered seawater in 1 nm steps by UV-Vis spectrophotometry (Varian Cary 100 Bio; Varian Inc, USA), using 10 cm path length quartz cuvettes pre-rinsed three times with ultra-pure water (Milli-Q: Millipore System Inc., USA). Spectra were blank corrected using Milli-Q water and for machine drift using the mean absorption from 700 nm to 800 nm. Measurements were made in triplicate and Naperian absorbance coefficients $a$ (m$^{-1}$) were determined following Hu et al. (2002). For each measurement the mean value and standard error of the mean is reported.

We chose not filter our CDOM samples based on our earlier work (Kitidis et al., 2006; Stubbins et al., 2006) that established strong relationships between CDOM in filtered and unfiltered seawater for coastal and oceanic waters. Filtration can lead to the contamination of dissolved organic carbon (DOC) and UV absorbance (Ferrari 2000, Karanfil et al., 2003 and Kitidis et al., 2006). Although our samples include both dissolved and particulate components of absorbance and are subject to scattering by particles that include living phytoplankton (Nelson and Siegel 2013 and references therein), any potential effects on our CDOM measurements can be considered minor relative to those likely to be introduced during filtration. Total CDOM absorbance was calculated as the integrated absorbance from 250 to 450 nm at a 1-nm resolution (Helms et al., 2008). The 250:365 nm absorption ratio ($E_2 : E_3$) was used to track relative changes in low molecular weight (LMW) vs high molecular weight (HMW) organic matter (OM); $E_2 : E_3$ decreases with increasing OM molecular size due to increasing light

absorption by HMW OM towards longer wavelengths (Peuravuori and Pihlaja, 1997). Spectral slope ($S$) was calculated using a nonlinear fit of an exponential function to the absorption spectrum over the ranges 275nm - 295nm and 350nm - 400nm (Helms et al., 2008). The spectral slope ratio, $S_R$ (= $S_{275-295}$ / $S_{350-400}$), was used to broadly characterize OM in terms of molecular weight and source; samples with low $S_R$ are of high molecular weight and have a greater tendency to be allochthonous (Helms et al., 2008). Due to instrument maintenance no CDOM measurements are available for the 17/07/13 survey.

The SSW samples (volume ~93 L) were used to estimate the spatial variability in $k_w$ using a fully automated, closed air–water gas exchange tank. Tank design, operation and routine procedures for its rigorous cleaning are all described in detail in Schneider-Zapp et al. (2014). In brief, the system generates water-side turbulence with an electronic baffle operated at three increasingly turbulent boundary conditions of 0.6, 0.7 and 0.75 Hz. Although turbulence created in a laboratory tank inevitably differs from turbulence *in-situ,* which is primarily wind-driven, our experimental system avoids the practical complications of simulating wind-induced turbulence in a laboratory while maintaining well-defined and reproducible conditions (Schneider-Zapp et al., 2014). The purpose of the tank is to facilitate an improved understanding of $k_w$ suppression by natural surfactants that will subsequently enable more complex experiments involving wind-driven turbulence. We used SSW in the tank experiments for two reasons. First, there is no practical procedure for collecting a large volume sample of surface seawater that preserves the integrity of the SML. Second, we have shown (i) that following its disturbance by vigorous mixing in a laboratory tank the SML becomes re-established on a time scale of seconds with respect to surfactants and other SML components (Cunliffe et al., 2013); (ii) that a new SML is similarly established when sub-surface coastal waters are pumped into large mesocosm tanks (Cunliffe et al., 2009).

The tank is coupled to two gas chromatographs (GC's) and an integral equilibrator in a continuous gas-tight loop (Schneider-Zapp et al., 2014). This enables temporal changes in the partial pressures of gaseous tracers artificially enriched in the tank water ($SF_6$, $CH_4$, and $N_2O$) to be measured simultaneously in tank water and headspace, thereby deriving three independent estimates of $k_w$ for each turbulence setting. Due to the dependence of $k_w$ on the Schmidt Number ($Sc$: the ratio of kinematic viscosity to gas diffusivity) raised to the power $n$ (Upstill-Goddard, 2006), the $k_w$ estimates were converted to $k_{660}$, the value of $k_w$ for $Sc$ = 660 (the value for $CO_2$ in seawater at 20 $^o$C), assuming $n$ = 0.5 for a wavy surface (Upstill-Goddard, 2006). As the work presented here pre-dates the installation of an analytical capability for $N_2O$ reported in Schneider-Zapp et al. (2014), our $k_{660}$ estimates are based on $CH_4$ and $SF_6$ only. To check for any biogenic changes to tracer concentrations, their loss via leakage or problems arising from GC analytical drift, the total masses of $CH_4$ and $SF_6$ were continually estimated over the duration of the experiments, from their measured partial pressures and the known water and air volumes (mass balance; Schneider-Zapp et al., 2014, Eq. 14). Experiments with a significant mass balance error (> ±5 % mass drift throughout the entire experiment) were excluded. For this study most of our $SF_6$ data failed the mass balance test because they fell outside the linear range of the GC detector. Consequently, we have excluded any analyses deriving from our $SF_6$ $k_w$ data, even though the resulting values of $k_{660}$ agree well with those obtained with $CH_4$.

The uncertainty in each $k_{660}$ measurement was derived via Gaussian error propagation (Schneider-Zapp et al., 2014; Tayler, 1996) and is typically smaller than ±0.8 cm hr$^{-1}$ (n = 48). We ascribe one result outside this range (±3.9 cm hr$^{-1}$) to salt crystal formation that we observed in the water equilibration circuit. To clarify the comparative SA effect on $k_w$ between sites we normalized our derived $k_{660}$ values to the value of $k_{660}$ derived in identical experiments in which the seawater samples were replaced by surfactant-free Milli-Q water (i.e. $R_{660}$ = $k_{660Sample}$ / $k_{660Milli-Q}$) (Schneider-Zapp et al., 2014).

**Results and Discussion**

Fig. 2 shows the spatial and temporal variability of SA, total CDOM absorbance from 250-450 nm ($CDOM_{250-450}$), $E_2 : E_3$ and $S_R$. For all four parameters temporal variability generally exceeded spatial variability, both in the SML and in SSW (Table S1 and S2). In both SA was highest during June-July (SML; 0.25 - 0.38 mg l$^{-1}$ T-X-100, n = 8, SSW; 0.09 - 0.28 mg l$^{-1}$ T-X-100, n = 8) and lowest during October-February (SML; 0.08 - 0.27 mg l$^{-1}$ T-X-100, n = 10, SSW; 0.09 - 0.19 mg l$^{-1}$, n = 10). SA was generally higher in the SML than in SSW, as previously observed (e.g. Wurl et al., 2011), and our values for both the SML and SSW are broadly consistent with those from an earlier study in this region of the coastal North Sea (Salter, 2010), although that work also reported an exceptionally high SA value (1.42 mg l$^{-1}$ T-X-100) coincident with a period of extreme river discharge that we did not experience. Our SA data also agree well with those obtained under non-slick conditions in the Santa Barbara Channel, California (Wurl et al., 2009), but they are at the low end of the range presented by Wurl et al. (2011) for the open ocean.

SA enrichment factors (EF = $C_{SML}$ / $C_{SSW}$; C = concentration) ranged between ~1.0 - 1.9, except for 2 samples collected in February 2013 for which SA was more enriched in SSW (EFs = 0.75 and 0.89, respectively). While in SSW there was an overall decrease in SA with increasing distance offshore, SA in the SML was more spatially and temporally variable (Fig. 2). Although two surveys (04/10/12; 10/06/13) clearly show an overall decrease in SA offshore the evidence is inconclusive because the remaining two transects shows either no clear overall trend (13/02/13) or are incomplete (17/07/13). These EFs are broadly consistent with values for the global ocean (Wurl et al., 2011) despite our SA values being lower overall. Importantly, there is a clear relationship between SA in the SML and SA in the SSW (r$^2$ = 0.81 p = <0.001 n = 18, $SA_{SSW}$ = 0.7664$SA_{SML}$ + 0.0183). This is supportive of the notion that SA in the SML is constantly renewed from the SSW (Cunliffe et al., 2013).

Total $CDOM_{250-450}$ ranged from 240 - 874 in the SML and 217 – 792 in the SSW (n = 15), ranges that are consistent with those for high salinity waters in the Chesapeake Bay (Helms et al., 2008). $CDOM_{250-450}$ decreased offshore by up to a factor ~ 4 (166 - 875) in both the SML and SSW. However, $CDOM_{250-450}$ was generally higher in the SML than in SSW (EFs = 0.8 – 1.8), similar to the case for SA. This is indicative of an accumulation of light absorbing components in the SML. Using the $E_2 : E_3$ and $S_R$ indices as tracers for OM composition we found that $E_2 : E_3$ was generally lower in the SML (2.46 - 6.83, n = 15) than in SSW (2.25 – 8.43, n = 15; EFs = 0.7 – 1.1), similar to $S_R$ (SML; 1.11 - 1.99, n = 15, SSW; 1.37 - 2.25, n = 15; EFs = 0.6 – 1.2). These ranges for both $E_2 : E_3$ and $S_R$ are consistent with values reported for the Chesapeake Bay (Helms et al., 2008). In our study $E_2 : E_3$ showed an opposite tendency to $CDOM_{250-450}$, progressively increasing seaward by up to a factor ~ 4 (2.3 - 8.43). In contrast, $S_R$ was generally higher in the SSW than in the SML and it exhibited both increases and decreases with distance offshore. Unlike SA, this latter behaviour characterized both the SML and SSW (Fig. 2). In October 2012 $S_R$ generally increased with distance offshore but in February and June 2013 it generally decreased offshore.

In earlier work $E_2 : E_3$ was considered largely independent of total CDOM absorbance (Helms et al., 2008); however, whilst we observed a weaker and less significant relationship between $CDOM_{250-450}$ and $E_2 : E_3$ in the SML (r$^2$ = 0.45, p = 0.06, n = 15), we found a strong relationship in SSW (r$^2$ = 0.69, p = <0.001, n = 15). In contrast, we found no consistent relationship for either $CDOM_{250-450}$ or $E_2 : E_3$ with $S_R$. We tentatively propose that the divergence we specifically observed between $E_2 : E_3$ and $S_R$ in February and June 2013 may be related to additional higher molecular weight organic matter of autochthonous origin during this period. However, there is no apparent relationship between Chl-a in the SSW (range 0.09 – 1.54 mg l$^{-1}$, n = 20; Table S2), which is a proxy for in-situ primary productivity (e.g. Frka et al., 2011), and either $CDOM_{250-450}$, $E_2 : E_3$ or $S_R$. Unequivocally establishing the underlying reasons for this requires additional surveys coupled with more advanced molecular characterization of OM (e.g. Lechtenfeld et al., 2013) and consideration of the potential roles of other light absorbing compounds (see review by Nelson and Siegel, 2013).

We similarly interpret the observed temporal variability in SA as a consequence of the mixing of SA sources related to at least two distinct marine and terrestrial endmembers. Likely candidates are terrestrially-derived SA from the nearby River Blyth and autochthonous SA from *in-situ* biological activity. The relatively high SA and $CDOM_{250-450}$ at site 1 (0 km, directly at the mouth of the river), which is a persistent feature of the data, is consistent with this explanation if it is assumed that the terrestrial SA endmember source is more abundant there than autochthonous derived SA. This scenario is supported by the lack of any clear relationship between SA and Chl-a. Furthermore, despite the covariance between $E_2 : E_3$ and $CDOM_{250-350}$, the overall decrease in $CDOM_{250-450}$ and increase $E_2 : E_3$ with distance offshore implies either dilution of terrestrially derived CDOM with lower molecular weight marine CDOM or photochemical degradation of higher molecular weight material. This is in agreement with other studies that showed either HMW CDOM breakdown by photochemical or microbial processes (e.g. Helms et al*., 2008; 2013) or an *in-situ* supply of LMW CDOM to the most seaward sites via primary productivity (i.e. lipid production; Frka et al., 2011). Either of these processes could explain the observed relationship between $E_2 : E_3$ and $CDOM_{250-450}$ but further work clarifying the dominant pathways of OM processing in our study area is required. As for total SA, these data reveal a distinction between the SML and SSW as previously observed (Frew et al., 2006; Wurl et al., 2009, 2013; Lechtenfeld et al., 2013; Cunliffe et al., 2013; Engel and Galgani 2016). The generally higher SSW values of $S_R$ and $E_2 : E_3$ noted earlier suggest that SML DOM is predominantly of HMW, as compared to predominantly LMW DOM in SSW. Although River Blyth discharge data [National River Flow Archive Centre for Ecology and Hydrology; http://www.ceh.ac.uk/data/nrfa/] show no clear relationship between river flow and either SA or CDOM at site 1, our observations of SA and $CDOM_{250-450}$ are broadly consistent with the results of a study in Cape Cod coastal waters (Frew et al., 2004), although in that study the offshore gradients in SA were much stronger than those we found off the NE UK coast. We also note that the River Blyth is rather small (mean annual discharge ~2 m$^3$ s$^{-1}$) and this is reflected in the very weak salinity gradients that we observed (range typically ±0.6).

The values we derived for $k_{660}$ using our gas exchange tank (Table S3: 6.82 to 22.06 cm hr$^{-1}$) are realistic in that they are within the reported natural $k_{660}$ range of 5.81 to 70.17 cm hr$^{-1}$ (Asher, 2009). Even so, it was not our intention to reproduce conditions leading to the generation of turbulence *in-situ*, which is in any case unachievable in a laboratory setting due to the multiple and variable controls of both water- and air-side turbulence (Upstill-Goddard, 2006). It is also important to reiterate that our approach was specifically designed to produce comparative $k_{660}$ estimates at controllable and reproducible turbulence levels and that as such, the mode of turbulence generation (water-side: motor driven baffle) is of secondary importance. Generally we found increasing $k_{660}$ with increasing water-side turbulence (Table S3). Even so, at the highest turbulence setting the data were comparatively noisy.

We found a strong correlation between $R_{660}$ in our tank experiments and SA in the SML *in-situ* (Fig. 3). Along with our observation of strong correlations between SA in the SML and in the SSW, this finding reinforces the notion of continual SML renewal from the underlying water (Hardy et al., 1982; Frew 2006 and Cunliffe et al., 2013). The strongest linear relationships between $R_{660}$ and SA *in-situ* in the SML were observed at 0.6 Hz (r$^2$ = 0.61, p = <0.001, n = 16) and 0.7 Hz (r$^2$ = 0.70, p = <0.001, n = 16). At 0.75 Hz the relationship was notably weaker (r$^2$ = 0.49, p = 0.02, n = 16) and only if a non-linear fit is applied to the data does the correlation coefficient improve. Although this could imply a threshold level of turbulence beyond which the surfactant effect on $k_{660}$ is rapidly attenuated *in-situ*, we think it more likely indicates the limitations of the method at high baffle speeds due to the formation of bubbles and/or wave breaking and aerosol generation. We observed a similar phenomenon with an earlier gas exchange tank design (Upstill-Goddard et al., 2003). Further data are required to distinguish between these possibilities and to more robustly establish the relationships between SA and $R_{660}$. We found no significant relationships between $R_{660}$ and either Chl-a, $CDOM_{250-450}$, $S_R$, $E_2 : E_3$ or other specific CDOM

wavelengths (e.g. $\lambda_{254}$, $\lambda_{300}$, $\lambda_{330}$, $\lambda_{350}$ or $\lambda_{443}$) *in-situ,* in either the SML or SSW. Given this evidence, the use of remotely sensed CDOM as predictor of $k_w$ variability over large spatial scales may well prove erroneous.

Our derived $R_{660}$ values (Fig. 3 and Table S3) correspond to $k_{660}$ suppressions of 14 - 51 % for our samples (Fig. 3; $R_{660}$ = 0.86 - 0.49, n = 48) and highlight the effect of $k_w$ suppression by natural surfactants in seawater as compared to surfactant-free water. We generally observed $k_w$ suppression to be higher (i.e., lowest $R_{660}$) at near-shore sites (< 5 km) than at off-shore sites (> 5 km). This is most clearly shown for $k_{660}$ derived for the lowest turbulence setting (0.6 Hz) in June 2013, for which $R_{660}$ at the most near-shore site was ~10 % lower than at the most offshore site. We also observed the largest seasonal

variance in $R_{660}$ at the most near-shore site; the range in $k_w$ suppression was 15 to 24 % (Fig. 3). Site 3 located 10 km offshore showed the highest variance in $R_{660}$ at the highest baffle setting but was consistently observed to show a high variability in $R_{660}$ between summer (June/July) and autumn/winter (October/February) regardless of changes in turbulence. The range of $k_w$ suppressions we observed (15 - 24 %) is within the range of previous laboratory work that demonstrated 10 - 90 % surfactant suppression of $O_2$ exchange for oceanic and coastal waters (Frew, 1997; Goldman et al., 1988) and $k_w$

reductions of 5 - 50 % for phytoplankton exudates (Frew et al., 1990). Similarly, laboratory and field experiments with artificial surfactants found 60 - 74 % $k_w$ suppression (Brockmann et al., 1982; Bock et al., 1999; Frew, 1997; Salter et al., 2011).

Given that our methodological approach was specifically designed to constrain the effect of surfactants in the SML on $k_w$ and
that this minimized the effects of other potential $k_w$ controls, our observations of distinct changes in the quantity and composition of OM in the SML and SSW prompt us to hypothesize that the observed spatio-temporal variation in $R_{660}$ and its relationship with SA (Fig. 3) is a consequence of compositional differences in the surfactant fraction of the SML DOM pool. The principal driver of this hypothesis is the data scatter inherent in the relationship between $R_{660}$ and SA. While we have not been able to unequivocally relate any control of $k_w$ to CDOM absorbance characteristics, and by inference CDOM
composition, we nevertheless hypothesise that a rigorous characterisation of the chemical composition of the surfactant pool will yield important insights into surfactant sources and biogeochemical processing that, when analysed in the context of physical forcing such as variable wind regime and hydrography (e.g. Chen et al., 2013; Frew et al., 2006; Gašparović et al., 2007; Lechtenfeld et al., 2013), will inform a better understanding of the spatial and temporal variability in $k_w$.

**Implications**

Understanding the physical and biogeochemical controls of air–sea gas exchange is necessary for establishing biogeochemical models for predicting regional- and global-scale trace gas fluxes and feedbacks. Our results demonstrate the potential of our gas exchange tank concept in providing important information of this nature. In estimating ocean $CO_2$ uptake for example, the spatio-temporal variability of $k_w$ is now a much larger uncertainty than the spatio-temporal variability of $p\text{CO}_2$ (Takahashi et al., 2009). For $CO_2$ and other climate-active gases that have strong sources and sinks in coastal waters
(e.g. $CH_4$, $N_2O$, halocarbons) it is equally important to quantify the degree of spatio-temporal variability in $k_w$ so as to better constrain regional trace gas budgets (Prowe et al., 2009; Thomas et al., 2004, 2007; Tsunogai et al., 1999). Clearly, this must involve resolving the influence on $k_w$ exerted not only by changes in total surfactant amount in the SML but also by variability in the composition of the surfactant pool and in the composition of the overall DOM pool. We anticipate that further seasonal measurements of the type described in this paper, both at coastal and ocean basin scales, will move us some
way towards an eventual full parameterization of the environmental controls of $k_w$, and in particular the evidently important roles played by spatial and temporal trends in both surfactant amount and composition.

**Acknowledgements**

This research was facilitated by grants from the Leverhulme Trust (RPG-303) and the UK Natural Environment Research Council (NE/IO15299/1) to RCU-G and from the German Research Foundation (DFG research fellowship) to KS-Z. We
also thank the crew of RV *Princess Royal* for field and logistical support, and Juliane Bischoff, Jon Barnes and David Whitaker for their technical advice and analytical assistance.

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

**Figure Captions**

Figure 1: Location map of the Dove Time Series sampling transect 2012-2013 off the coast of Blyth, North East England.


Figure 2: Surfactant activity (normalized to Triton T-X-100), CDOM (Total $a$ 250-450 nm), $S_R$ and $E_2 : E_3$ ratio of the sea surface microlayer (SML; top panels) and sub-surface water (SSW; bottom panels) from a North Sea transect during 2012-2013.

Figure 3: Right panels are $R_{660}$ for $CH_4$ in seawater samples collected along the Dove Time Series transect in the North Sea during 2012-2013. Left panels are scatterplots of the $R_{660}$ ($R_{660} = k_{660Sample} / k_{660Milli-Q}$) and surfactant activity (SA) in the SML. The top two windows are for a tank baffle frequency of 0.6 Hz, the two middle windows are for a tank baffle frequency 0.7 Hz and bottom two windows are for a tank baffle frequency of 0.75 Hz.

**Supplement**

Table S1: Surfactant activity (normalized to Triton T-X-100), total CDOM absorbance (250-450 nm), $S_{275-295}$, $S_{350-400}$, $S_R$, $E_2 : E_3$ ratio and SA enrichment factor (EF) of the sea surface microlayer (SML) from a North Sea transect during 2012-2013.

| Date | Dove Station | SML SA mg l$^{-1}$ T-X-100 | SML CDOM$_{250-450}$ Total $a$ | SML $S_{275-295}$ nm$^{-1}$ | SML $S_{350-400}$ nm$^{-1}$ | SML $S_R$ | SML $E_2 : E_3$ | SA EF |
|---|---|---|---|---|---|---|---|---|
| 04/10/12 | 1 | 0.27 ±0.03 | 874.8 ±37.6 | 0.013 ±0.001 | 0.010 ±0.001 | 1.32 ±0.01 | 3.580 ±0.250 | 1.44 ±0.40 |
| 04/10/12 | 2 | 0.22 ±0.08 | 429.0 ±156.7 | 0.016 ±0.002 | 0.008 ±0.004 | 1.99 ±0.01 | 4.291 ±1.026 | 1.90 ±0.71 |
| 04/10/12 | 3 | 0.17 ±0.03 | 240.4 ±6.8 | 0.021 ±0.001 | 0.014 ±0.001 | 1.54 ±0.01 | 6.832 ±0.208 | 1.5 ±0.28 |
| 04/10/12 | 4 | 0.15 ±0.02 | 250.6 ±12.8 | 0.021 ±0.001 | 0.013 ±0.001 | 1.62 ±0.01 | 6.584 ±0.177 | 1.33 ±0.20 |
| 04/10/12 | 5 | 0.14 ±0.02 | 261.1 ±33.3 | 0.020 ±0.001 | 0.012 ±0.001 | 1.73 ±0.01 | 6.009 ±0.385 | 1.19 ±0.21 |
| 13/02/13 | 1 | 0.11 ±0.01 | 637 ±158.5 | 0.010 ±0.002 | 0.005 ±0.003 | 1.91 ±0.01 | 2.459 ±0.561 | 0.89 ±0.12 |
| 13/02/13 | 2 | 0.08 ±0.01 | 339.1 ±12.8 | 0.014 ±0.001 | 0.008 ±0.001 | 1.74 ±0.01 | 3.847 ±0.062 | 0.75 ±0.17 |
| 13/02/13 | 3 | 0.13 ±0.03 | 452.1 ±12.2 | 0.013 ±0.001 | 0.009 ±0.001 | 1.51 ±0.01 | 3.556 ±0.132 | 1.28 ±0.34 |
| 13/02/13 | 4 | 0.10 ±0.01 | 326.0 ±23.8 | 0.016 ±0.001 | 0.009 ±0.001 | 1.71 ±0.01 | 4.179 ±0.205 | 1.11 ±0.22 |
| 13/02/13 | 5 | 0.11 ±0.03 | 322.6 ±14.3 | 0.016 ±0.001 | 0.011 ±0.001 | 1.47 ±0.01 | 4.236 ±0.109 | 1.04 ±0.36 |
| 10/06/13 | 1 | 0.30 ±0.03 | 379.0 ±10.8 | 0.020 ±0.001 | 0.011 ±0.001 | 1.85 ±0.01 | 5.849 ±0.124 | 1.08 ±0.13 |
| 10/06/13 | 2 | 0.29 ±0.01 | 409.8 ±41.9 | 0.019 ±0.001 | 0.011 ±0.001 | 1.74 ±0.01 | 5.640 ±0.206 | 1.07 ±0.04 |
| 10/06/13 | 3 | 0.30 ±0.01 | 328.5 ±11.7 | 0.020 ±0.001 | 0.012 ±0.001 | 1.71 ±0.01 | 6.188 ±0.086 | 1.08 ±0.07 |
| 10/06/13 | 4 | 0.28 ±0.01 | 400.5 ±10.5 | 0.016 ±0.001 | 0.015 ±0.001 | 1.11 ±0.01 | 5.162 ±0.055 | 1.09 ±0.05 |
| 10/06/13 | 5 | 0.25 ±0.01 | 326.9 ±13.2 | 0.018 ±0.001 | 0.014 ±0.001 | 1.29 ±0.01 | 5.832 ±0.067 | 1.05 ±0.02 |
| 17/07/13 | 1 | 0.38 ±0.04 | | | | | | 1.51 ±0.18 |
| 17/07/13 | 2 | 0.25 ±0.05 | | | | | | 1.00 ±0.20 |
| 17/07/13 | 3 | 0.28 ±0.05 | | | | | | 1.09 ±0.18 |
| 17/07/13 | 4 | | | | | | | |
| 17/07/13 | 5 | | | | | | | |


Table S2: Surfactant activity (normalized to Triton T-X-100), total CDOM absorbance (250-450 nm), $S_{275-295}$, $S_{350-400}$, $S_R$, $E_2 : E_3$ ratio, salinity and Chl-a of sub-surface water (SSW) from a North Sea transect during 2012-2013.

| Date | Dove Station | SA<br>mg l$^{-1}$ T-X-100 | CDOM$_{250-450}$<br>Total $a$ | $S_{275-295}$<br>nm$^{-1}$ | $S_{350-400}$<br>nm$^{-1}$ | $S_R$ | $E_2 : E_3$ | Chl-a<br>mg L$^{-1}$ | Salinity |
|---|---|---|---|---|---|---|---|---|---|
| 04/10/12 | 1 | 0.19 ±0.05 | 530.5 ±0.7 | 0.015 ±0.001 | 0.011 ±0.001 | 1.37 ±0.01 | 4.21 ±0.04 | 0.55 ±0.03 | 33.1 |
| 04/10/12 | 2 | 0.12 ±0.01 | 241.5 ±5.5 | 0.020 ±0.001 | 0.013 ±0.001 | 1.59 ±0.01 | 6.44 ±0.19 | 0.53 ±0.01 | 34.0 |
| 04/10/12 | 3 | 0.12 ±0.01 | 240.0 ±3.5 | 0.021 ±0.001 | 0.013 ±0.001 | 1.60 ±0.01 | 6.66 ±0.08 | 1.05 ±0.02 | 34.0 |
| 04/10/12 | 4 | 0.11 ±0.01 | 227.2 ±3.8 | 0.021 ±0.001 | 0.013 ±0.001 | 1.63 ±0.01 | 6.60 ±0.11 | 1.54 ±0.04 | 34.2 |
| 04/10/12 | 5 | 0.12 ±0.01 | 220.3 ±0.8 | 0.021 ±0.001 | 0.013 ±0.001 | 1.65 ±0.01 | 6.74 ±0.08 | 1.00 ±0.02 | 34.1 |
| 13/02/13 | 1 | 0.12 ±0.01 | 792.2 ±10.7 | 0.009 ±0.001 | 0.005 ±0.001 | 1.80 ±0.01 | 2.25 ±0.10 | 0.35 ±0.02 | 33.8 |
| 13/02/13 | 2 | 0.10 ±0.02 | 376.0 ±34.3 | 0.014 ±0.001 | 0.007 ±0.001 | 1.93 ±0.01 | 3.42 ±0.25 | 0.15 ±0.02 | 33.9 |
| 13/02/13 | 3 | 0.10 ±0.01 | 478.3 ±9.9 | 0.013 ±0.001 | 0.008 ±0.001 | 1.65 ±0.01 | 3.30 ±0.01 | 0.14 ±0.01 | 33.5 |
| 13/02/13 | 4 | 0.09 ±0.02 | 377.1 ±24.4 | 0.017 ±0.001 | 0.008 ±0.001 | 2.14 ±0.01 | 3.95 ±0.03 | 0.09 ±0.03 | 33.9 |
| 13/02/13 | 5 | 0.11 ±0.03 | 292.4 ±2.3 | 0.016 ±0.001 | 0.009 ±0.001 | 1.80 ±0.01 | 4.32 ±0.10 | 0.11 ±0.01 | 33.7 |
| 10/06/13 | 1 | 0.27 ±0.01 | 322.8 ±2.1 | 0.020 ±0.001 | 0.009 ±0.001 | 2.25 ±0.01 | 5.93 ±0.04 | 0.65 ±0.04 | 33.6 |
| 10/06/13 | 2 | 0.28 ±0.01 | 284.1 ±3.2 | 0.021 ±0.001 | 0.010 ±0.001 | 2.20 ±0.01 | 6.55 ±0.08 | 0.61 ±0.09 | 34.1 |
| 10/06/13 | 3 | 0.28 ±0.01 | 261.9 ±0.8 | 0.022 ±0.001 | 0.011 ±0.001 | 2.02 ±0.01 | 7.05 ±0.06 | 0.36 ±0.02 | 34.0 |
| 10/06/13 | 4 | 0.26 ±0.01 | 253.8 ±1.0 | 0.021 ±0.001 | 0.011 ±0.001 | 1.95 ±0.01 | 7.26 ±0.11 | 0.33 ±0.01 | 34.2 |
| 10/06/13 | 5 | 0.24 ±0.01 | 217.2 ±1.2 | 0.023 ±0.001 | 0.013 ±0.001 | 1.76 ±0.01 | 8.43 ±0.14 | 0.22 ±0.07 | 34.2 |
| 17/07/13 | 1 | 0.25 ±0.01 | | | | | | 0.12 ±0.10 | 34.6 |
| 17/07/13 | 2 | 0.25 ±0.01 | | | | | | 0.35 ±0.06 | 34.4 |
| 17/07/13 | 3 | 0.25 ±0.01 | | | | | | 0.38 ±0.08 | 34.3 |
| 17/07/13 | 4 | | | | | | | 0.12 ±0.01 | 34.5 |
| 17/07/13 | 5 | | | | | | | 0.14 ±0.01 | 34.6 |


Table S3: $k_{660}$ and R$_{660}$ estimates for three water-side turbulence settings of 0.6, 0.7 and 0.75 Hz from North East coast transect 2012-2013

| Date | Dove Station | CH$_4$ $k_{660}$ | | | CH$_4$ $R_{660}$ | | |
|------|------|--------|--------|--------|--------|--------|--------|
| | | 0.6 Hz | 0.7 Hz | 0.75 Hz | 0.6 Hz | 0.7 Hz | 0.75 Hz |
| 04/10/12 | 1 | 9.58 ±0.52 | 11.21 ±0.45 | 11.40 ±0.28 | 0.60 ±0.03 | 0.70 ±0.03 | 0.72 ±0.02 |
| 04/10/12 | 2 | 9.39 ±0.48 | 11.51 ±0.46 | 11.02 ±0.26 | 0.59 ±0.03 | 0.72 ±0.03 | 0.69 ±0.02 |
| 04/10/12 | 3 | 9.58 ±0.52 | 12.62 ±0.52 | 11.25 ±0.29 | 0.60 ±0.03 | 0.79 ±0.03 | 0.71 ±0.02 |
| 04/10/12 | 4 | 10.50 ±0.40 | 12.32 ±0.49 | 10.66 ±0.31 | 0.66 ±0.02 | 0.77 ±0.03 | 0.67 ±0.02 |
| 04/10/12 | 5 | 12.25 ±0.55 | 13.59 ±0.57 | 12.77 ±0.34 | 0.77 ±0.03 | 0.85 ±0.04 | 0.80 ±0.02 |
| 13/02/13 | 1 | 12.17 ±0.56 | 12.79 ±0.55 | 12.72 ±0.35 | 0.76 ±0.04 | 0.80 ±0.03 | 0.80 ±0.02 |
| 13/02/13 | 2 | | | | | | |
| 13/02/13 | 3 | 12.68 ±0.66 | 13.67 ±0.56 | 11.16 ±0.42 | 0.80 ±0.04 | 0.86 ±0.04 | 0.70 ±0.03 |
| 13/02/13 | 4 | 10.97 ±0.46 | 13.21 ±0.53 | 12.55 ±0.34 | 0.69 ±0.03 | 0.83 ±0.03 | 0.79 ±0.02 |
| 13/02/13 | 5 | 13.04 ±0.51 | 13.6 ±0.54 | 13.11 ±0.37 | 0.82 ±0.03 | 0.85 ±0.03 | 0.82 ±0.02 |
| 10/06/13 | 1 | 8.17 ±0.37 | 9.05 ±0.40 | 8.70 ±0.44 | 0.51 ±0.02 | 0.57 ±0.02 | 0.55 ±0.03 |
| 10/06/13 | 2 | 7.77 ±0.33 | 8.93 ±0.37 | 8.65 ±0.25 | 0.49 ±0.02 | 0.56 ±0.02 | 0.54 ±0.02 |
| 10/06/13 | 3 | 8.02 ±0.33 | 10.46 ±0.43 | 11.54 ±0.42 | 0.50 ±0.02 | 0.66 ±0.03 | 0.72 ±0.03 |
| 10/06/13 | 4 | | | | | | |
| 10/06/13 | 5 | 9.51 ±0.38 | 10.52 ±0.44 | 9.83 ±0.26 | 0.60 ±0.02 | 0.66 ±0.03 | 0.62 ±0.02 |
| 17/07/13 | 1 | 9.71 ±0.41 | 10.92 ±0.47 | 10.13 ±0.36 | 0.61 ±0.03 | 0.69 ±0.03 | 0.64 ±0.02 |
| 17/07/13 | 2 | 10.29 ±0.54 | 9.94 ±0.47 | 9.50 ±0.24 | 0.65 ±0.03 | 0.62 ±0.03 | 0.60 ±0.02 |
| 17/07/13 | 3 | 10.02 ±0.62 | 11.15 ±0.48 | 7.91 ±3.90 | 0.63 ±0.04 | 0.70 ±0.03 | 0.50 ±0.02 |
| 17/07/13 | 4 | | | | | | |
| 17/07/13 | 5 | | | | | | |



Figure 1

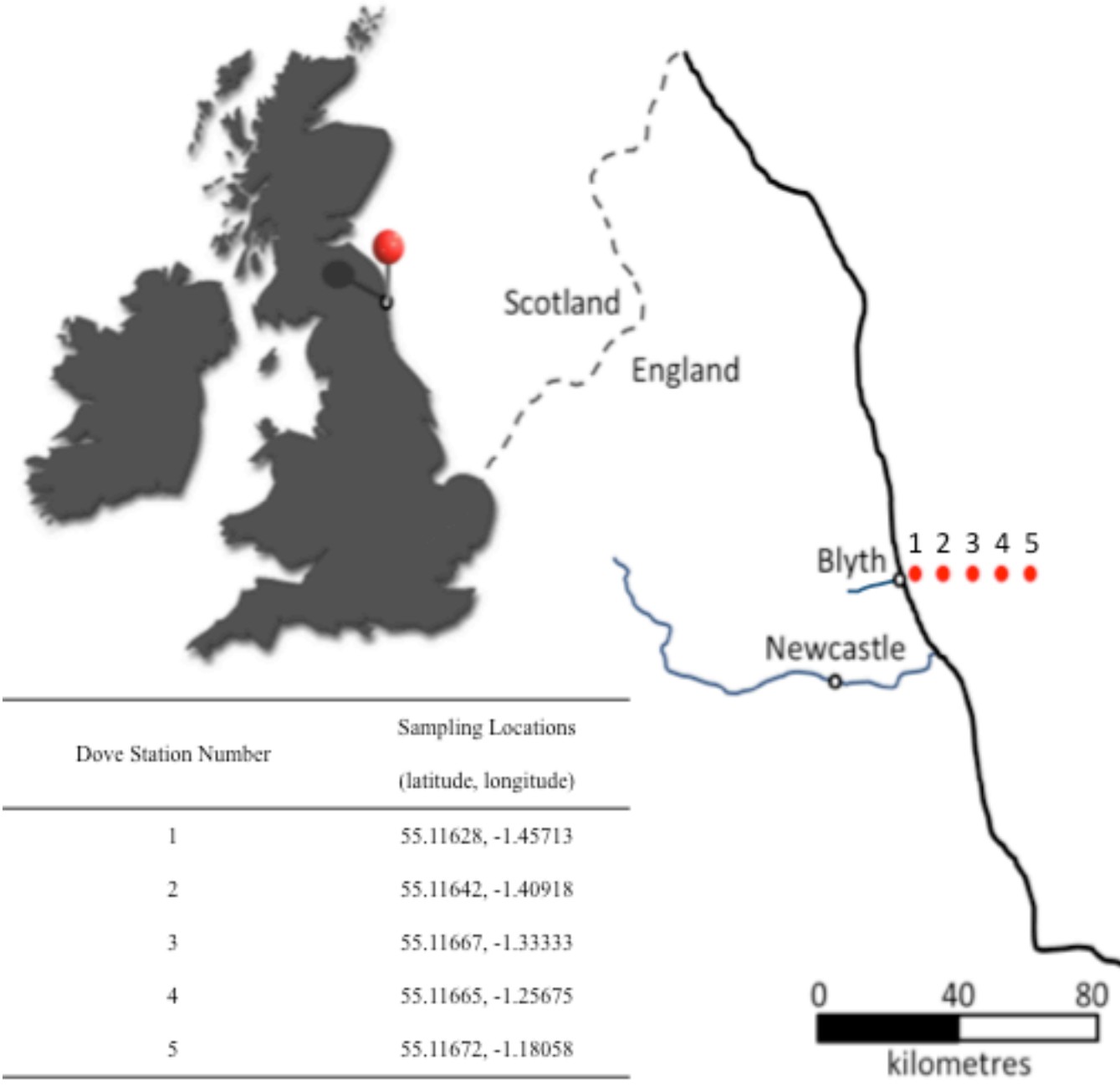

| Dove Station Number | Sampling Locations (latitude, longitude) |
|---|---|
| 1 | 55.11628, -1.45713 |
| 2 | 55.11642, -1.40918 |
| 3 | 55.11667, -1.33333 |
| 4 | 55.11665, -1.25675 |
| 5 | 55.11672, -1.18058 |

Figure 2

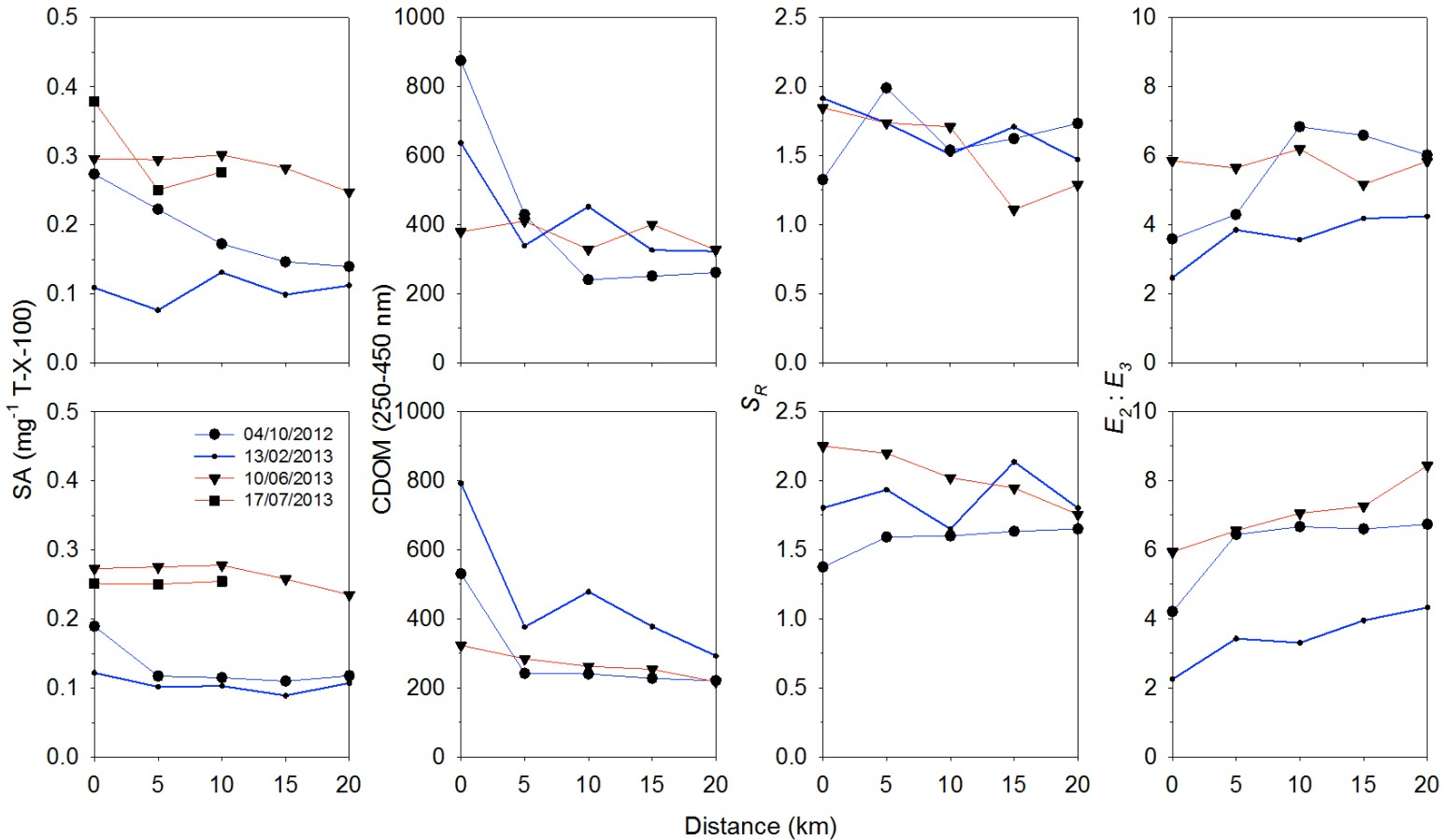


Figure 3

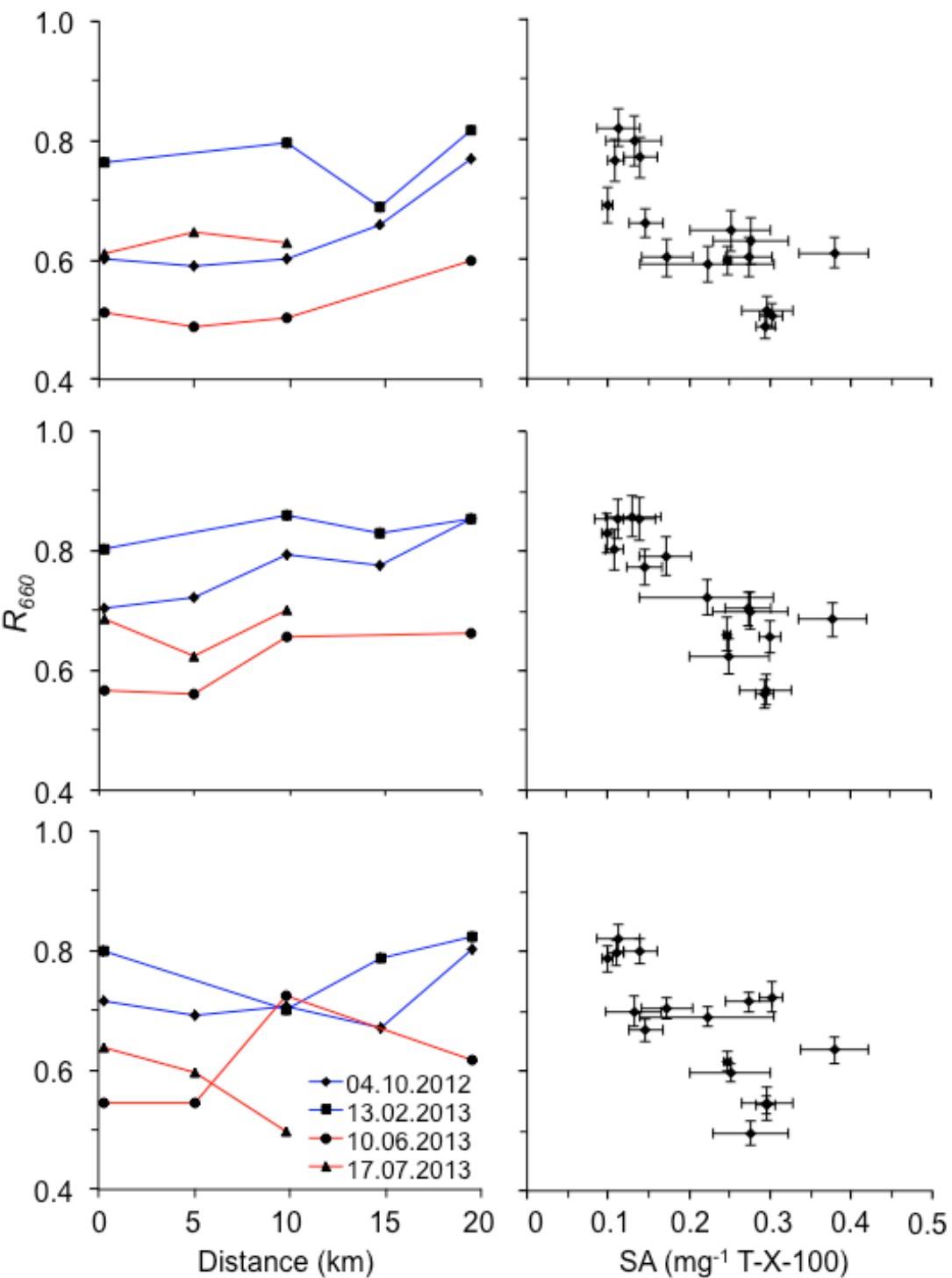