# Peer review of "Surfactant control of gas transfer velocity along an offshore coastal transect: results from a laboratory gas exchange tank"

_Biogeosciences, 2016_

## Referee Comment (RC1) · Anonymous Referee #1 · 15 Feb 2016

Review on ms. Bg-2016-7: "Surfactant control of gas transfer velocity along an off-shore coastal transect: results from a laboratory gas exchange tank" by R. Pereira, K. Schneider-Zapp and R. C. Upstill-Goddard

General comments: The manuscript deals with a current and interesting topic targeting to contribute to the understanding of surfactant control on gas transfer velocity across air-sea boundary. This may add to the knowledge on the role of oceans in the climate changes. The author collected sea surface microlayer and subsurface water and measured surfactant activity and coloured dissolved organic matter. Using a custom-designed air-sea gas exchange tank the authors evaluated corresponding values of the gas transfer velocity. To my opinion Abstract and Results and discussion section

should be rewritten. Abstract reads as compiled Result section. It should be rewritten or expanded with the main conclusions. Results and discussion also give a lot of results with too short discussion. The aims are poorly written. Therefore I suggest major revision along comments listed below. Specific comments: Abstract: P1, l 15: for nonprofessional photochemist it is not clear what does it mean: k660 (kw for CO2; freshwater; 20oC). Is it CO2 or CH4? P1, l 19-20: I do not understand how such variability should be taken into account when evaluating marine trace gas sources.

Introduction p1, l 32: surfactants are organics as well p 2, l 41-43: The aims are poorly written and must be rewritten. The aims should be written as hypotheses which are tested in the paper. Why CDOM was measured? Part of aims was written on p3, l 80-81.

Materials and Methods p2, l 48: I am not sure if triplicate sampling is necessary as sampling is time consuming while both SML and SSW are not in a steady-state. p2, l 64: Are CDOM measurements performed in the filtered samples? If not which is the influence of the particles (living and non-living) on the measured data? p3, l 79: Applied turbulence settings should be listed here.

Results and Discussion p3 l 111: While the authors state here: "For all four parameters temporal variability generally exceeded spatial variability" in the Abstract the statement is opposite: "Spatial SA variability exceeded its temporal variability."??? Those are opposite statements. P4, l 116: If the authors suggest that there was expected relationship between SA in the SML and SSW, it should be cited. p4, L 136: I do not understand why the authors explicitly discuss DOM as SML is always enriched in POM of non-living and living origin. Are the measurements carried out on filtered samples? It should be pointed out in the Methods section. p4, l 144: Surfactant activity of autochtonous origin can be very high during bloom period. The authors may check Chl a data for sampling dates from the satellite observations. P4, l 146: I suggest saying lower molecular weight marine CDOM, than LMW marine CDOM. The authors do not know on molecular weight of marine CDOM, apart from well-known fact that terrestrial

humics are of higher molecular weight than marine humics. P 6, L 200-202: The authors state: "the observed spatio-temporal variation in R660 and its relationship with SA (Fig. 3) is a consequence of compositional differences in the surfactant fraction of the SML DOM pool". Unfortunately throughout Results and Discussion section it is not discussed in such way. I suggest improving this section by discussing straightforward the influence of surfactant having different composition on R660. p6, l 222: Pity that the authors did not measured DOC. These measurements would give information on the hydrobicity/hydrophilicity of surfactants (high SA and low DOC-hydrophobic surfactants; high SA and high DOC-hydrophilic surfactants).

Technical corrections: The authors should be consistent in writing: sub-surface water or subsurface water P1, l 7: I suggest adding full name of CDOM P 2, l 46-47: I suggest removing Table 1 and adding coordinates into Fig. 1. p5, l 184: Twice written "at the" p5, l188: October is Autumn and not Winter. Table 2: I suggest adding borders between different sampling dates to allow easier data comparison. It should be defined what is total CDOM absorbance (250-450 nm). Is it integration or is it average value over the whole 250-450 nm spectra? The same data are given both in the Table 2 and Fig. 2, with more data (S 279-295, S 350-400, Salinity) given in Table 2. I suggest that the authors decide on how to present data, in figure or table. Personally I prefer data given in Figs. than in tables. Similar situation is with Table 3 and Fig. 3. Table 2: S275–295, S350–400 and salinity are not listed in the table caption. Table 3. Three different turbulence settings should be listed in the caption. p4, l 121: comma to be removed Figure 1. It would be good to draw Blyth River into this fig. Figures: I suggest lines and symbols to be presented in different colours for sampling dates. This would significantly increase the clearness.
* * *

---

## Referee Comment (RC2) · Anonymous Referee #2 · 3 Mar 2016

R. Pereira et al. present data on the effect of surfactants on the gas transfer velocity between the ocean and atmosphere. This subject is poorly understood, but essential for the understanding of air-sea gas exchange of climate-relevant gases. The study utilizes a laboratory gas exchange tank designed by the same group. The tank allows comparing gas transfer velocity under artificial turbulence, but under in situ turbulence as acknowledged by the authors. It still allows investigating fundamental processes. However, the discussion of the results has to be elaborated, also in terms of the literature.

However, some corrections and suggestions are designated for the authors' attention; otherwise I recommend the manuscript as suitable for publication. Overall, prior publication some sections of the manuscripts have to be rewritten, especially section of Results and Discussion. For example, the authors need to discuss their observations with the available literature on the chemical composition of the SML.

Abstract I agree with the comments of reviewer #1 that the abstract is too much based on results without outlining the main conclusions.

Introduction L35, P1: Under slick conditions, e.g. wave-damped water surfaces, surfactant activities are probably high enough to form a dense layer acting as a barrier. However, under non-slick conditions, still under influence of surfactants, I believe passing of gases through the air-sea interface is controlled by slow diffusion-driven transfer, not because the interface is a barrier.

I also feel that the introduction misses to describe biological properties of the air-sea interface (or sea surface microlayer) as microbes could be a direct source of surfactants.

The authors miss also to describe the purpose and aim of the study. It should be added as a final paragraph to this section describing overall objectives and hypothesis.

Materials and Methods L46, P2: I assume for the field work a medium-sized vessel was required, and I am wondering how the authors can be sure to collect the SML free of disturbance and contamination inevitably caused by the vessel. L61, P2: Not clear if CDOM was measured in both SML and SSW, or only in SSW. How were the samples filtered? L82, P3: What is the range of applied turbulence? How is it measured? As TKE? L84, P3: I got the meaning, but an odd sentence hard to grasp.

Results and Discussion P111, P3: Temporal versus spatial variability are poorly presented and discussed, and, as also mentioned by Reviewer #1, opposite statement appears in abstract. Reference to a figure would make it clearer. P115, P3: Provide regression plot. P value exact 0.001? As expected from . . .citation? P116, P3: provide standard deviation for the range of EF.

P122, P4: How do the authors define "high spatial and temporal variability"? Giving the range, doesn't provide any information about variability. P133,P4: The relationship for SML is not strong (r2=0.45) and even not significant (p=0.06). This misinterpretation have to be corrected. P137,P4: Lichterfeld et al. 2013 report finding about HMW DOM in SML. Have to be discussed here. P151, P4: There is plenty of literature available (Lichterfeld et al., 2013, work by Carlson, work by Frew) supporting the observation. Also known enrichment of TEP supports the idea of a HMW matrix in the SML (see Wurl et al. and Cunliffe et al.)

---

## Referee Comment (RC3) · Anonymous Referee #3 · 3 Mar 2016

General comments:

The authors present results on surfactant control over gas transfer velocities across the sea-air interface. These results arise from seasonal field sampling of sea-surface microlayer and subsurface water along an offshore coastal transect combined with laboratory experiments using a custom-designed air-sea gas exchange tank (Schneider-Zapp et al., 2014). The comparison of field and laboratory study was applied to derive $k_w$ from natural samples. The field parameters analysed were surfactant activity (SA) and coloured dissolved organic matter (CDOM). In the laboratory, $CH_4$ partial pressure was monitored to derive gas transfer velocity $k_{660}$ ($k_w$ for $CO_2$ in seawater). The authors conclude that surfactant activity in the SML can lead to a $k_{660}$ suppression

between 14 and 51% with strong seasonal and spatial gradients in coastal waters.

The topic presented here is indeed very interesting and of great importance for ocean and atmospheric scientists, and can make a significant contribution to the present understanding of oceanic control over atmospheric gases concentration. However, despite the relevance of the topic and its link to climate change, the aims of the study as well as potential biological implications are poorly presented. Since the study deals with biogenically organics produced, I would expect some more discussion in this respect that would expand the perception of this work across disciplines. I also think that the authors should mention how their study contributes to the understanding of air-sea gas exchange in relation to expanding oxygen minimum zones and ocean acidification, in the introduction as well as in the implications section. As an example, a recent special issue "Biogeochemical processes, tropospheric chemistry and interactions across the ocean–atmosphere interface in the coastal upwelling off Peru" (BG/ACP/AMT/OS inter-journal SI) deals with trace gases emission from coastal upwelling regimes characterized by high biological productivity. It is a totally different marine system but worth mentioning. I think the discussion shall be rewritten. It is hard to follow the authors' argumentation, especially concerning CDOM. Results of SA and CDOM should be better linked to the gas-exchange tank experiment. Finally, I think some sentences in the introduction should refer to Eddy covariance measurements of sea-air fluxes of gases such as $CO_2$ as comparison to the presented data and technique.

Based on these considerations, the manuscript needs major improvement; therefore I would suggest publication after major revisions.

Specific comments:

Abstract: I agree with the first referee that abstract should be expanded, and I suggest including the aims of the study. Page 1, line 14: please revise. There is not enough information to state that terrestrially-derived CDOM can be biogeochemically processed in North-Sea coastal waters. Page 1, line 15: k660 isn't it for $CO_2$ in seawater (and not

freshwater?)

Introduction: Page 1, line 24: which sources and sinks? Please be more specific Page 1, line 25: which are the "environmental controls" for CO2? Please specify. Please make a short introduction to the sea-surface microlayer and the nanolayer (e.g. Lass et al. 2013, Biogeosciences, 10, 5325–5334) as at line 35, page 1 you refer to the "monolayer". Page 2, lines 38-40: please specify what are natural surfactants and explain why you choose to use CDOM as a tracer in this particular study, and what has been previously shown for CDOM in the SML. Also, some more details are needed in describing "procedural difficulties".

Methods and results: Since you sampled a transect of the Dove Time Series, I suppose there should be more parameters available to describe the physical and biological environment. It would be useful to have temperature, wind speed, DOC, chlorophyll data, as an example. Also, what was the estimated thickness of the SML you collected? How did you avoid ship-contamination while sampling from the RV Princess Royal? Please shortly introduce the enrichment factor. In the results you present a mean EF. I think a median EF is more suitable than a mean EF to avoid excess weighting of a few high-enriched samples that shift EFs towards higher values. Page 2, line 66: why did you use the mean value for CDOM (250-450 nm)? Please explain. I suppose (from table 2) that the mean value is calculated between 250 and 450 nm, please expand the method section. How were CDOM samples treated? Did you filter through GF/F, or PES 0.45 $\mu$m..? For how long were the samples stored before analysis? Are these samples the same from Schneider-Zapp et al. 2013? If so, make a specific reference. Page 3, lines 84-86: it is a long sentence, please rephrase. Page 3, line 11: which one was higher, spatial or temporal variability? This sentence is opposite to the abstract. Line 112: please give some reference values for SA in the SML from the literature. Page 4, line 1: why do you expect a relationship between SML and SSW? Please explain. A variation in EF from 1 to 1.9 means no enrichment or relatively high enrichment. Are these mean/min/max values? It is hard to understand EFs is no error estimation on EF

SSW, which has been shown in many studies, and make references to those. Line 203: please give examples with the aid of the literature of what kind of organic components, both from biological and anthropogenic sources, can be part of the surfactant fraction of the SML.

---

## Author Comment (AC1) · 13 Apr 2016

*Ryan Pereira*
*email: r.pereira@hw.ac.uk*
*Tel:+44 (0)131 451 3537*

**The Sir Charles Lyell Centre**
**Heriot-Watt University**
**Edinburgh**
**EH14 4AS**

13th April 2016

Response to the review comments of manuscript "**Surfactant control of gas transfer velocity along an offshore coastal transect: results from a laboratory gas exchange tank**" Ref: bg-2016-7.

Dear Dr Herndl,

Thank you for accepting the invitation to be editor on our manuscript. In response to the email dated 16 March 2016 advising that the open discussion of our manuscript has been closed I am please to provide you with a point-by-point response to the reviewer comments. We would like to state that we greatly appreciated the reviewers' support of our research and we found their comments extremely helpful and constructive.

We note that the procedure in the email dated 16 March 2016 advised us to provide a detailed response to the reviewer comments before resubmitting our manuscript. However, given that all the reviewers were supportive of our research and scientific approach but suggested changes to the text of the manuscript we found it necessary to revise the manuscript to adequately address the reviewer comments and suggestions. We have been advised by the *Biogeosciences* editorial team not to upload this revised version on to the Discussion platform and have included excerpts of the revised manuscript in our response letter for you consideration. If however, you would like to see the revised version of the manuscript this can be made available to you upon request.

We hope that you we find our approach suitable and beneficial in assessing our manuscript for publication in *Biogeosciences*. If you require any further information please contact me.

Yours sincerely,

Dr Ryan Pereira (corresponding author) on behalf of, Klaus Schneider-Zapp and Robert Upstill-Goddard.

**Response to Reviewer 1**

*1.    The manuscript deals with a current and interesting topic targeting to contribute to the understanding of surfactant control on gas transfer velocity across air-sea boundary. This may add to the knowledge on the role of oceans in the climate changes.*

Reply: We thank the reviewer for their positive outlook and interest in our research.

*2.    To my opinion Abstract and Results and discussion section should be rewritten. Abstract reads as compiled Result section. It should be rewritten or expanded with the main conclusions.*

Reply: Following the reviewers general suggestions and more specific comments detailed below we have revised the Abstract and the Results and Discussion Section. We have expanded the Abstract to introduce the importance of the topic from lines 9-12 *"Understanding the physical and biogeochemical controls of air–sea gas exchange is necessary for establishing biogeochemical models for predicting regional- and global-scale trace gas fluxes and feedbacks.  To this end we report the results of experiments designed to constrain the effect of surfactants in the sea surface microlayer (SML) on the gas transfer velocity ($k_w$; cm hr$^{-1}$), seasonally (2012-2013) along a 20 km coastal transect (North East UK)."* and added a section at the end from lines 22-26 *"Total CDOM absorbance (250-450 nm), the CDOM spectral slope ratio ($S_R = S_{275–295} / S_{350–400}$), the 250:365 nm CDOM absorption ratio ($E_2 : E_3$) and Chl-a all indicated spatial and temporal signals in the quantity and composition of organic matter in the SML and SSW. This prompts us to hypothesize that spatio-temporal variation in $R_{660}$ and its relationship with SA is a consequence of compositional differences in the surfactant fraction of the SML DOM pool that warrants further investigation.".*

*3.    Results and discussion also give a lot of results with too short discussion. The aims are poorly written. Therefore I suggest major revision along comments listed below.*

Reply: We thank the reviewer for their constructive and useful comments on our manuscript. Given your generally positive review, support of our experimental design and general interpretation of our results we do not believe that the science manuscript requires major revision and that the comments relate mainly to the text of the original manuscript. Accordingly, we have modified the text adding a clear aims paragraph *"The overarching goal of this study was to derive a fundamental understanding of the spatial and temporal control of $k_w$ variability by surfactant. Our testable hypothesis is that due to surfactant suppression of $k_w$, inverse correlations between $k_w$ and surfactant activity (SA) in the SML should temporally persist in regions where SA shows high spatial variability.  A secondary aim was to ascertain whether surfactant accumulation in the SML is strongly linked to primary productivity (using chlorophyll-a as a proxy) and whether CDOM could be used as a quantitative index of SA and $k_w$, given its widespread use in remote sensing platforms."*(lines 70-75) in the Introduction and further discussion around our results where appropriate (detailed below and highlighted in the marked revision of our manuscript).

*4.    Abstract: P1, l 15: for nonprofessional photochemist it is not clear what does it mean: k660 (kw for CO2; freshwater; 20$^o$C). Is it CO2 or CH4?*

Reply: The *k660* refers to the gas transfer velocity normalised to the gas transfer velocity of $CO_2$ at 20°C in seawater (please note that 'freshwater' was in error and should have read as seawater as pointed out by Reviewer 2; Point 54, which is now corrected in the revised manuscript). To clarify this we have revised the abstract to read "$k_{660}$ *($k_w$ for $CO_2$ in seawater at 20 °C)*" (line 17) and added more details on $k_{660}$ in the methods section (lines 148-152).

*5.        P1, l 19-20: I do not understand how such variability should be taken into account when evaluating marine trace gas sources.*

Reply: The aim of our study was to demonstrate that surfactants present either naturally or anthropogenically in coastal systems have an effect on gas transfer velocities. The $k_w$ variability with SA in situ that we observe is a first step that needs to be explored further including incorporation into various models.

*6.        Introduction p1, l 32: surfactants are organics as well*

Reply: We agree and have added "other" prior to organics (line 37).

*7.        p 2, l 41-43: The aims are poorly written and must be rewritten. The aims should be written as hypotheses which are tested in the paper.*

Reply: We completely agree (see response to Point 3). In the original version of the manuscript the aims were scattered throughout the text. They have now been collated in the modified text adding a clear aims paragraph (lines 70-75), with our hypothesis to be tested in the Introduction.

*8.        Why CDOM was measured?*

Reply: CDOM was used in our study as it is a powerful tool to provide semi-quantitative information on the composition of organic matter with aqueous environments Furthermore we wanted to examine whether *"CDOM could be used as a quantitative index of SA and $k_w$, given its widespread use in remote sensing platforms"* which is now stated in lines 72-75).

*9.        Part of aims was written on p3, l 80-81.*

Reply: See response to Point 3 and 7. This line has now been relocated to the Introduction.

*10.       Materials and Methods p2, l 48: I am not sure if triplicate sampling is necessary as sampling is time consuming while both SML and SSW are not in a steady-state.*

Reply: Triplicate sampling was conducted to ensure both sample and analytical reproducibility. Whilst this does add more time on return to the laboratory it gave us more confidence in the results produced from our analysis.

*11.    p2, l 64: Are CDOM measurements performed in the filtered samples? If not which is the influence of the particles (living and non-living) on the measured data?*

Reply: Our CDOM measurements were made on unfiltered seawater. We have clarified this in the Methods section by stating, *"We chose not filter our CDOM samples based on our earlier work (Kitidis et al., 2006; Stubbins et al., 2006) that established strong relationships between CDOM in filtered and unfiltered seawater for coastal and oceanic waters. Filtration can lead to the contamination of dissolved organic carbon (DOC) and UV absorbance (Ferrari 2000, Karanfil et al., 2003 and Kitidis et al., 2006). Although our samples include both dissolved and particulate components of absorbance and are subject to scattering by particles that include living phytoplankton (Nelson and Siegel 2013 and references therein), any potential effects on our CDOM measurements can be considered minor relative to those likely to be introduced during filtration."* (lines 115-120).

*12.    p3, l 79: Applied turbulence settings should be listed here.*

Reply: The turbulence settings have been added to the Methods section (lines 133-134).

*13.    Results and Discussion p3 l 111: While the authors state here: "For all four parameters temporal variability generally exceeded spatial variability" in the Abstract the statement is opposite: "Spatial SA variability exceeded its temporal variability."??? Those are opposite statements.*

Reply: We apologize for this error in the Abstract and have now corrected this in the revised text. The Abstract now reads *"Temporal SA variability exceeded its spatial variability."* (lines 14-15).

*14.    P4, l 116: If the authors suggest that there was expected relationship between SA in the SML and SSW, it should be cited.*

Reply: We have now expanded this section of the manuscript adding detail to SA enrichment factors and included the relationship observed between SA in the SML and SSW to now read as *"These EFs are broadly consistent with values for the global ocean (Wurl et al., 2011) despite our SA values being lower overall. Importantly, there is a clear relationship between SA in the SML and SA in the SSW ($r^2$ = 0.81 p = <0.001 n = 18, $SA_{SSW}$ = 0.7664$SA_{SML}$ + 0.0183). This is supportive of the notion that SA in the SML is constantly renewed from the SSW (Cunliffe et al., 2013)."* (lines 181 – 185).

*15.    p4, L 136: I do not understand why the authors explicitly discuss DOM as SML is always enriched in POM of non-living and living origin. Are the measurements carried out on filtered samples? It should be pointed out in the Methods section.*

Reply: We have revised "DOM" to now read as "OM" to include both dissolved and particulate fractions. We have also added more CDOM details to the Methods section (see reply to Point 11).

*16.    p4, l 144: Surfactant activity of autochtonous origin can be very high during bloom period. The authors may check Chl a data for sampling dates from the satellite observations.*

Reply: We have added our own Chl-a dataset as recommended by the reviewer. We have added this to the Methods section (Lines 104-107), and the Results and Discussion section (Lines 204-208). Intriguingly we find no relationship between SA and Chl-a, which does not preclude that SA is autochthonously produced but suggests that Chl-a is not a good indicator for SA production in our study.

*17.     P4, l 146: I suggest saying lower molecular weight marine CDOM, than LMW marine CDOM. The authors do not know on molecular weight of marine CDOM, apart from well-known fact that terrestrial humics are of higher molecular weight than marine humics.*

Reply: We agree and this has been changed in the revised manuscript *"...implies either dilution of terrestrially derived CDOM with lower molecular weight marine CDOM or photochemical degradation of higher molecular weight material."* (line 216-217).

*18.     P 6, L 200-202: The authors state: "the observed spatio-temporal variation in R660 and its relationship with SA (Fig. 3) is a consequence of compositional differences in the surfactant fraction of the SML DOM pool". Unfortunately throughout Results and Discussion section it is not discussed in such way. I suggest improving this section by discussing straightforward the influence of surfactant having different composition on R660.*

Reply: As previously mentioned we have added more discussion around our SA, CDOM and Chl-a results (lines 187-195, 203-208, 241-243 and 249-253). We have also added further explanation of this hypothesis to read *"The principal driver of this hypothesis is the data scatter inherent in the relationship between $R_{660}$ and SA. While we have not been able to unequivocally relate any control of $k_w$ to CDOM absorbance characteristics, and by inference CDOM composition, we nevertheless hypothesise that a rigorous characterisation of the chemical composition of the surfactant pool will yield important insights into surfactant sources and biogeochemical processing that, when analysed in the context of physical forcing such as variable wind regime and hydrography (e.g. Chen et al., 2013; Frew et al., 2006; Gasparovic et al., 2007; Lechtenfeld et al., 2013), will inform a better understanding of the spatial and temporal variability in $k_w$."* in the revised manuscript (lines 273-278).

*19.     p6, l 222: Pity that the authors did not measured DOC. These measurements would give information on the hydrophobicity/hydrophilicity of surfactants (high SA and low DOC-hydrophobic surfactants; high SA and high DOC-hydrophilic surfactants).*

Reply: We agree that it would useful to have DOC measurements but unfortunately this was not part of the sampling plan. In future studies we will include these measurements.

*20.     Technical corrections: The authors should be consistent in writing: sub-surface water or subsurface water*

Reply: This is now consistent throughout out the manuscript to read as "*sub-surface*".

*21.     P1, l 7: I suggest adding full name of CDOM P 2, l 46-47*

Reply: Done

*22.    I suggest removing Table 1 and adding coordinates into Fig. 1. p5, l 184*

Reply: Done. The sample locations are now shown in Fig. 1.

*23.    Twice written "at the" p5, l188*

Reply: This has been corrected.

*24.    October is Autumn and not Winter.*

Reply: We agree and has been changed to *"Autumn/Winter"* (Line 262).

*25.    Table 2: I suggest adding borders between different sampling dates to allow easier data comparison. It should be defined what is total CDOM absorbance (250-450 nm). Is it integration or is it average value over the whole 250-450 nm spectra?*

Reply: We have added borderlines where appropriate to conform to EGU *Biogeosciences* guidelines. We have fully defined the CDOM 250-450nm measurement in the Methods section to read as *"Total CDOM absorbance was calculated as the integrated absorbance from 250 to 450 nm at a 1-nm resolution (Helms et al., 2008)."* (Lines 120-122).

*26.    The same data are given both in the Table 2 and Fig. 2, with more data (S 279-295, S 350-400, Salinity) given in Table 2. I suggest that the authors decide on how to present data, in figure or table. Personally I prefer data given in Figs. than in tables. Similar situation is with Table 3 and Fig. 3.*

Reply: We chose to include all of our data in tables to for future access by any reader but felt that figures better show and describe our results for discussion. However, noting the reviewers suggestion we have included the full datasets presented in this manuscript in the auxiliary material.

*27.    Table 2: S275–295, S350–400 and salinity are not listed in the table caption. Table 3. Three different turbulence settings should be listed in the caption.*

Reply: This has now been changed but is this Table is now Table S3.

*28.    p4, l 121: comma to be removed Figure 1. It would be good to draw Blyth River into this fig.*

Reply: Done.

*29.    Figures: I suggest lines and symbols to be presented in different colours for sampling dates. This would significantly increase the clearness.*

Reply: Done.

Response to Reviewer 2

*30.      R. Pereira et al. present data on the effect of surfactants on the gas transfer velocity between the ocean and atmosphere. This subject is poorly understood, but essential for the understanding of air-sea gas exchange of climate-relevant gases. The study utilizes a laboratory gas exchange tank designed by the same group. The tank allows comparing gas transfer velocity under artificial turbulence, but under in situ turbulence as acknowledged by the authors. It still allows investigating fundamental processes. The discussion of the results has to be elaborated, also in terms of the literature. However, some corrections and suggestions are designated for the authors' attention; otherwise I recommend the manuscript as suitable for publication.*

Reply: We thank the reviewer for his/her constructive comments on our manuscript and support of our research study. As stated in response to Point 3 we have expanded the Introduction and Results and Discussion sections to include relevant literature.

*31.      Overall, prior publication some sections of the manuscripts have to be rewritten, especially section of Results and Discussion. For example, the authors need to discuss their observations with the available literature on the chemical composition of the SML.*

Reply: We have revised the Results and Discussion section to include comparisons to relevant literature in Lines 169-175, 181-194 and 217-223. Given that we do not have explicit results to discuss the overall chemical composition of the SML in our study (as we only use semi-quantitative analyses) we detail the general composition of the SML and SNL in the Introduction (Lines 40-59). This now reads as "*The surface ocean boundary with the atmosphere is characterised by the sea surface microlayer (SML) which is ~ 400 µm or less deep and is physically and biogeochemically distinct from the underlying water (Cunliffe et al., 2013). Dissolved components and buoyant particles from the underlying water become enriched in the SML by bubble scavenging (Cunliffe at al., 2009; Cunliffe et al., 2013; Gaŝparović et al., 1998; Petrović et al., 2002; Wurl et al., 2011 and Źutić et al., 1981), leading to accelerated rates of microbiological and photochemical processes (Cunliffe et al., 2013; (Vodacek et al., 1997); Häder et al., 2011). Material accumulating in the SML includes a range of surface active substances (surfactants) such as transparent exopolymer particles (TEP; Wurl and Holmes, 2008), polysaccharides (Sieburth et al., 1976), lipid-like material (Gasporavic et al 1998; Kattner and Brockmann 1978; Lass and Friedrichs, 2011), amino acids (Kuznetsova et al., 2004) and chromophoric dissolved organic matter (CDOM; Tilstone et al., 2010). The tendency is for many of these components to be of lower molecular weight than their analogues in the underlying water (Lechtenfeld et al., 2013) and this may be coupled to in situ primary production (Chin et al., 1998; Passow 2002), allochthonous inputs of terrestrial material of either natural (e.g. Frew et al., 2006) or anthropogenic (Guitart et al., 2007) origin, and the photochemical and/or microbial reworking of higher molecular weight material (Tilstone et al., 2010; Schulz et al., 2013).*".

*32.      Abstract I agree with the comments of reviewer #1 that the abstract is too much based on results without outlining the main conclusions.*

Reply: We have revised the manuscript abstract to reflect more of the research importance and our main conclusions. (see response to Point 2).

*33.	Introduction L35, P1: Under slick conditions, e.g. wave-damped water surfaces, surfactant activities are probably high enough to form a dense layer acting as a barrier. However, under non-slick conditions, still under influence of surfactants, I believe passing of gases through the air-sea interface is controlled by slow diffusion-driven transfer, not because the interface is a barrier.*

Reply: As suggested by Reviewer 3 we have expanded the Introduction to include the sea surface nanolayer (see response to Point 56 and Lines 54-59). We have also revised the text with the aim of clarifying the possible influences of slick conditions to now read *"Its physico-chemical properties differ from those of the SML, providing an additional diffusion barrier and modifying the viscoelasticity of the air-sea interface (McKenna and Bock, 2006). This reduces the rate of air-sea gas exchange by wave damping and by attenuating turbulent energy transfer (Liss and Duce, 1997). It is these effects that are manifested in reductions in the value of $k_w$ (McKenna and McGillis, 2004; Salter et al., 2011)"* (lines 55-59).

*34.	I also feel that the introduction misses to describe biological properties of the air-sea interface (or sea surface microlayer) as microbes could be a direct source of surfactants.*

Reply: We agree that the manuscript would benefit by having an introduction on the biological properties of the SML and have now included this (lines 40-52; see response to Point 31).

*35.	The authors miss also to describe the purpose and aim of the study. It should be added as a final paragraph to this section describing overall objectives and hypothesis.*

Reply: This has now been changed. Please see response to Point 7.

*36.	Materials and Methods L46, P2: I assume for the field work a medium-sized vessel was required, and I am wondering how the authors can be sure to collect the SML free of disturbance and contamination inevitably caused by the vessel.*

Reply: The *RV Princess Royal* is a relatively small research vessel with the deck only 2-3 metres above the water surface. We made every effort to ensure that SML samples were collected from undisturbed areas of water and free of contamination following procedures outlined by Cunliffe et al., (2013). To clarify this in the main text we have revised the Methods section (Lines 81-84) to now read as *"At each station the SML and underlying sub-surface water (SSW) were sampled in triplicate. To minimise contamination from RV Princess Royal all samples were collected from near the bow whilst stationary and with the bow positioned upwind. Visual inspection for potential fouling from the research vessel prior to sampling aimed to ensure collection of a representative sample.".*

*37.	L61, P2: Not clear if CDOM was measured in both SML and SSW, or only in SSW. How were the samples filtered?*

Reply: We have clarified this in the revised manuscript. Please see the response to Point 11.

*38.     L82, P3: What is the range of applied turbulence? How is it measured? As TKE?*

Reply: The water-side turbulence is parameterized by the baffle speed used of 0.6, 0.7 and 0.75 Hz. We have slightly modified this in the revised text to clarify this (Lines 133-134).

*39.     L84, P3: I got the meaning, but an odd sentence hard to grasp.*

Reply: We have revised this sentence and expanded to clarify our point, which now reads as *"In brief, the system generates water-side turbulence with an electronic baffle operated at three increasingly turbulent boundary conditions of 0.6, 0.7 and 0.75 Hz. Although turbulence created in a laboratory tank inevitably differs from turbulence in situ, which is primarily wind-driven, our experimental system avoids the practical complications of simulating wind-induced turbulence in a laboratory while maintaining well-defined and reproducible conditions (Schneider-Zapp et al., 2014)."* (Lines 133-137).

*40.     Results and Discussion P111, P3: Temporal versus spatial variability are poorly presented and discussed, and, as also mentioned by Reviewer #1, opposite statement appears in abstract. Reference to a figure would make it clearer.*

Reply: We have revised the manuscript to discuss the spatial and temporal variability observed in our results in the context with other literature where appropriate. We have also referred to the appropriate figure (Lines 167, 241 and 255).

*41.     P115, P3: Provide regression plot. P value exact 0.001? As expected from . . .citation?*

Reply: We have revised this statement to include the relationship observed and add the "<" sign to now read as *"Importantly, there is a clear relationship between SA in the SML and SA in the SSW ($r^2$ = 0.81 p = <0.001 n = 18, $SA_{SSW}$ = 0.7664$SA_{SML}$ + 0.0183)"* (Lines 183-184). The data for this relationship is available in the additional data tables.

*42.     P116, P3: provide standard deviation for the range of EF.*

Reply: The standard deviations are included in the Table S1.

*43.     P122, P4: How do the authors define "high spatial and temporal variability"? Giving the range, doesn't provide any information about variability.*

Reply: We agree and have now discussed our seasonal and spatial variability in the context of other studies where appropriate throughout the Results and Discussion section.

*44.     P133, P4: The relationship for SML is not strong ($r2$=0.45) and even not significant (p=0.06). This misinterpretation have to be corrected.*

Reply: We have slightly modified this statement to now read as "*In earlier work $E_2$: $E_3$ was considered largely independent of total CDOM absorbance (Helms et al., 2008); however, whilst we observed a weaker and less significant relationship between $CDOM_{250-450}$ and $E_2$: $E_3$ in the SML ($r^2$ = 0.45, p = 0.06, n = 15) we found a strong relationship in SSW ($r^2$ = 0.69, p = <0.001, n = 15)*" (Lines 199-201).

*45.     P137,P4: Lichterfeld et al. 2013 report finding about HMW DOM in SML. Have to be discussed here.*

Reply: We agree and to clarify this in the revised manuscript we have added the following discussion "*We tentatively propose that the divergence we specifically observed between $E_2$: $E_3$ and $S_R$ in February and June 2013 may be related to additional higher molecular weight organic matter of autochthonous origin during this period. However, there is no apparent relationship between Chl-a in the SSW (range 0.09 – 1.54 mg $l^{-1}$, n = 20; Table S2), which is a proxy for in-situ primary productivity (e.g. Frka et al., 2011), and either $CDOM_{250-450}$, $E_2$ : $E_3$ or $S_R$. Unequivocally establishing the underlying reasons for this requires additional surveys coupled with more advanced molecular characterization of OM (e.g. Lechtenfeld et al., 2013) and consideration of the potential roles of other light absorbing compounds (see review by Nelson and Siegel 2013).*" (lines 202-208). Please note we believe that the Reviewer means the Lechtenfeld et al., 2013 paper not "Lichterfeld et al., 2013.

*46.     P151, P4: There is plenty of literature available (Lichterfeld et al., 2013, work by Carlson, work by Frew) supporting the observation. Also known enrichment of TEP supports the idea of a HMW matrix in the SML (see Wurl et al. and Cunliffe et al.)*

Reply: We have added further discussion to our revised manuscript about the accumulation of HMW OM in the SML from lines 217-223 to now read "*This is in agreement with other studies that showed either HMW CDOM breakdown by photochemical or microbial processes (e.g. Helms et al., 2008; 2013) or an in situ supply of LMW CDOM to the most seaward sites via primary productivity (i.e. lipid production; Frka et al., 2011). Either of these processes could explain the observed relationship between $E_2$ : $E_3$ and $CDOM_{250-450}$ but further work clarifying the dominant pathways of OM processing in our study area is required. As for total SA, these data reveal a distinction between the SML and SSW as previously observed (Frew et al., 2006; Wurl et al., 2009, 2013; Lechtenfeld et al., 2013; Cunliffe et al., 2013; Engel and Galgani 2016).*".

**Response to Reviewer 3**

*47.    The authors present results on surfactant control over gas transfer velocities across the sea-air interface. These results arise from seasonal field sampling of sea-surface microlayer and subsurface water along an offshore coastal transect combined with laboratory experiments using a custom-designed air-sea gas exchange tank (Schneider- Zapp et al., 2014). The comparison of field and laboratory study was applied to derive kw from natural samples. The field parameters analysed were surfactant activity (SA) and coloured dissolved organic matter (CDOM). In the laboratory, CH4 partial pressure was monitored to derive gas transfer velocity k660 (kw for CO2 in seawater). The authors conclude that surfactant activity in the SML can lead to a k660 suppression between 14 and 51% with strong seasonal and spatial gradients in coastal waters. The topic presented here is indeed very interesting and of great importance for ocean and atmospheric scientists, and can make a significant contribution to the present understanding of oceanic control over atmospheric gases concentration.*

Reply: We thank the reviewer for his/her comments and acknowledgment of the importance of our research.

*48.    However, despite the relevance of the topic and its link to climate change, the aims of the study as well as potential biological implications are poorly presented. Since the study deals with biogenically organics produced, I would expect some more discussion in this respect that would expand the perception of this work across disciplines.*

Reply: We have revised the manuscript to clearly state our aims and hypothesis (see response to Point 3). We have also revised the manuscript to present further discussion in the Introduction and around our results (see response to Point 3, 46 and 47).

*49.    I also think that the authors should mention how their study contributes to the understanding of air-sea gas exchange in relation to expanding oxygen minimum zones and ocean acidification, in the introduction as well as in the implications section. As an example, a recent special issue "Biogeochemical processes, tropospheric chemistry and interactions across the ocean–atmosphere interface in the coastal upwelling off Peru" (BG/ACP/AMT/OS inter-journal SI) deals with trace gases emission from coastal upwelling regimes characterized by high biological productivity. It is a totally different marine system but worth mentioning.*

Reply: Whilst we agree that in the 'big picture' of air-sea gas exchange OMZ and OA are important they are not topics relevant to the discussion of our study. We have no evidence to discuss these topics and feel that establishing the effect of gas transfer suppression by surfactants present in the SML and SSW is a big research question and study area in itself. This would also seem to be the opinion of all of the reviewers who are largely positive of our study (see Points 1 and 30 and 47). We have referenced the Peru study in the Results and Discussion section in the revised manuscript (line 223).

*50.    I think the discussion shall be rewritten. It is hard to follow the authors' argumentation, especially concerning CDOM. Results of SA and CDOM should be better linked to the gas-exchange tank experiment. Finally, I think some sentences in the*

*introduction should refer to Eddy covariance measurements of sea-air fluxes of gases such as CO2 as comparison to the presented data and technique.*

Reply: We have modified the text in the revised manuscript to make the manuscript easier to read and follow in our opinion. This is highlighted in the revised manuscript. However, we do not believe that comparing our data to Eddy covariance studies are appropriate in this case as the approaches to we employ are not comparable. We use a technique that produces a comparative number, not an absolute.

*51.     Based on these considerations, the manuscript needs major improvement; therefore I would suggest publication after major revisions.*

Reply: We thank the reviewer for their support in publication of our manuscript after the suggested revisions. We hope that the revised manuscript meets with your expectations.

*52.     Abstract: I agree with the first referee that abstract should be expanded, and I suggest including the aims of the study.*

Reply: We agree. Please see response to Point 2.

*53.     Page 1, line 14: please revise. There is not enough information to state that terrestrially-derived CDOM can be biogeochemically processed in North-Sea coastal waters.*

Reply: Please see response to Point 2.

*54.     Page 1, line 15: k660 isn't it for CO2 in seawater (and not freshwater?)*

Reply: We apologize for this error, which has now been corrected. (Please see Point 2 and 4).

*55.     Introduction: Page 1, line 24: which sources and sinks? Please be more specific Page 1, line 25: which are the "environmental controls" for CO2? Please specify.*

Reply: We have modified this sentence to now read *"The global budgets of important climate active gases such as carbon dioxide ($CO_2$), nitrous oxide ($N_2O$) and methane ($CH_4$) have important marine components that are predicted to change in a future climate (Bakker et al., 2014)."* (Lines 29-30) as we do not wish to replicate the message from the excellent publication by Bakker et al., (2014).

*56.     Please make a short introduction to the sea-surface microlayer and the nanolayer (e.g. Lass et al. 2013, Biogeosciences, 10, 5325–5334) as at line 35, page 1 you refer to the "monolayer".*

Reply: We agree that adding a section on the nanolayer is a useful addition to the manuscript and has been added from lines 54-59 to now read as *"The SML is itself overlain by the surface nanolayer (SNL); this is ~1–10 nm thick and can be a monolayer, it also comprises of surface-active substances and it may be enriched in carbohydrates*

*during summer (Lass et al., 2013). Its physico-chemical properties differ from those of the SML, providing an additional diffusion barrier and modifying the viscoelasticity of the air-sea interface (McKenna and Bock, 2006). This reduces the rate of air-sea gas exchange by wave damping and by attenuating turbulent energy transfer (Liss and Duce, 1997). It is these effects that are manifested in reductions in the value of $k_w$ (McKenna and McGillis, 2004; Salter et al., 2011).".*

*57. Page 2, lines 38-40: please specify what are natural surfactants and explain why you choose to use CDOM as a tracer in this particular study, and what has been previously shown for CDOM in the SML. Also, some more details are needed in describing "procedural difficulties".*

Reply: We have added more background to the SML composition in the Introduction of the revised manuscript (Lines-39-51). We have also modified our original statement '…procedural difficulties' to now read *"The role of natural surfactants has remained inadequately quantified due to the complexity of measuring $k_w$ in-situ and the spatial and temporal variability in natural surfactant concentration and composition (Salter et al., 2011)."* (Lines 62-64).

*58. Methods and results: Since you sampled a transect of the Dove Time Series, I suppose there should be more parameters available to describe the physical and biological environment. It would be useful to have temperature, wind speed, DOC, chlorophyll data, as an example.*

Reply: The DTS aim is to monitor long-term changes in plankton community changes in the North Sea, which we opportunistically sampled. This is now stated on lines 78-80. We have added a Chl-a dataset to our manuscript (see response to Point 16) but we do not have DOC measurements (see response to Point 19). We agree that directly measured wind speed would have been useful but this equipment was not present on the RV Princess Royal. In the absence of directly measured wind speed we utilized the ECMWF ERA Interim dataset (Lines 91-93). However, we could only use this information to briefly describe our study area as the spatial resolution of the dataset was not suitable for comparison with our study sites.

*59. Also, what was the estimated thickness of the SML you collected?*

Reply: The estimated sampling depth of the SML in 65-80 μm. We have added this information from lines 84-87 in the revised manuscript.

*60. How did you avoid ship-contamination while sampling from the RV Princess Royal?*

Reply: Please see response to Point 36.

*61. Please shortly introduce the enrichment factor. In the results you present a mean EF. I think a median EF is more suitable than a mean EF to avoid excess weighting of a few high- enriched samples that shift EFs towards higher values.*

Reply: We now briefly introduce EF on line 177. We believe there is some confusion by the reviewer as we present the mean EF from each of the sample replicates with the associated standard deviation presented in Table S1.

*62.    Page 2, line 66: why did you use the mean value for CDOM (250-450 nm)? Please explain. I suppose (from table 2) that the mean value is calculated between 250 and 450 nm, please expand the method section.*

Reply: Again we believe there is some confusion here. We utilized the integrated UV-Vis absorbance of our water samples from 250-450nm following the method of Helms et al., (2008) to demonstrate the spatial and temporal variation in CDOM concentrations in our study. The mean value again refers to the fact that we measured replicate samples from each site. The standard deviation of each mean measurement is presented in Table S1 and S2. We have revised the manuscript to clarify this (lines 120-122).

*63.    How were CDOM samples treated? Did you filter through GF/F, or PES 0.45 µm..? For how long were the samples stored before analysis?*

Reply: Please see response to Point 11. Samples were analyzed within 12 hours now stated on lines 97-98.

*64.    Are these samples the same from Schneider-Zapp et al. 2013? If so, make a specific reference.*

Reply: No these samples were not used in the technical note by Schneider-Zapp et al. 2013

*65.    Page 3, lines 84-86: it is a long sentence, please rephrase.*

Reply: Done. Please see response to Point 39.

*66.    Page 3, line 11: which one was higher, spatial or temporal variability? This sentence is opposite to the abstract.*

Reply: We apologize for this error, which has been correct in the revised version of the manuscript. Please see response to Point 4 and 54.

*67.    Line 112: please give some reference values for SA in the SML from the literature.*

Reply: We have added literature values for SA on lines 169-175 in the revised manuscript to read as *"SA was generally higher in the SML than in SSW, as previously observed (e.g. Wurl et al., 2011), and our values for both the SML and SSW are broadly consistent with those from an earlier study in this region of the coastal North Sea (Salter, 2010), although that work also reported an exceptionally high SA value (1.42 mg l$^{-1}$ T-X-100) coincident with a period of extreme river discharge that we did not experience. Our SA data also agree well with those obtained under non-slick conditions in the Santa Barbara Channel, California (Wurl et al., 2009), but they are at the low end of the range presented by Wurl et al. (2011) for the open ocean.".*

*68.	Page 4, line 1: why do you expect a relationship between SML and SSW? Please explain. A variation in EF from 1 to 1.9 means no enrichment or relatively high enrichment. Are these mean/min/max values? It is hard to understand EFs is no error estimation on EF is given, either. For example, assuming a 10% error on EFs calculation, would you consider EF = 1.1 being enriched in the SML and EF = 0.9 or 0.89 (for SA, for instance) being rather enriched in the SSW? You could apply the Gaussian error propagation analysis to EFs as well to obtain a clearer enrichment/depletion.*

Reply: Please see the response to Point 41 and 61 as we have now revised this section of the manuscript.

*69.	Page 4, line 123: I would first describe CDOM and then SR, as it derives from CDOM. Please give some reference values for SR. What does a number close to 1 or to 2 means? Compare to the literature.*

Reply: We have revised the manuscript to first describe total $CDOM_{250-450}$ before discussing the ratio indices (lines 186-194). We introduced the values of the CDOM ratios in the Methods section of the original manuscript (now stated on lines 122-128) and use our results to inter compare the quality of OM observed (lines 194-208).

*70.	Page 4, line 135: please specify what do you mean by marine endmembers.*

Reply: we have revised this statement to now read *"We tentatively propose that the divergence we specifically observed between $E_2 : E_3$ and $S_R$ in February and June 2013 may be related to additional higher molecular weight organic matter of autochthonous origin during this period."* on lines 202-204.

*71.	Line 146: why dilution with marine LMW-DOM? Could it just be a photochemical degradation over HMW-DOM terrestrially derived?*

Reply: We believe that both mechanisms are possible and should be investigated in future studies. We have revised our manuscript to now read as follows *"The relatively high SA and $CDOM_{250-450}$ at site 1 (0 km, directly at the mouth of the river), which is a persistent feature of the data, is consistent with this explanation if it is assumed that the terrestrial SA endmember source is more abundant there than autochthonous derived SA. This scenario is supported by the lack of any clear relationship between SA and Chl-a. Furthermore, despite the covariance between $E_2 : E_3$ and $CDOM_{250-350}$, the overall decrease in $CDOM_{250-450}$ and increase $E_2 : E_3$ with distance offshore implies either dilution of terrestrially derived CDOM with lower molecular weight marine CDOM or photochemical degradation of higher molecular weight material. This is in agreement with other studies that showed either HMW CDOM breakdown by photochemical or microbial processes (e.g. Helms et al., 2008; 2013) or an in situ supply of LMW CDOM to the most seaward sites via primary productivity (i.e. lipid production; Frka et al., 2011). Either of these processes could explain the observed relationship between $E_2 : E_3$ and $CDOM_{250-450}$ but further work clarifying the dominant pathways of OM processing in our study area is required. As for total SA, these data reveal a distinction between the SML and SSW as previously observed (Frew et al., 2006; Wurl et al., 2009, 2013; Lechtenfeld et al., 2013; Cunliffe et al., 2013; Engel and Galgani 2016)."* (lines 212-223).

*72.     Line 148: please specify why an in-situ supply of LMW DOM should take place. By what process? I think more data are needed to support this hypothesis, as well as more reference to the existing literature.*

Reply: Please see response to Point 71 above.

*73.     Lines 149-151: please specify what do you mean by clear compositional distinction between SML and SSW. Only in terms of MW or also of origin? Autochthonous marine DOM can be HMW-DOM as well. If no data on biological production are presented, it is hard to support this statement.*

Reply: The compositional distinction we observe is based on the CDOM datasets presented, which indicates different contributions of HMW and LMW OM in the SML and SSW. The aim of compositional statement is to discuss the possible sources of OM that may explain our observations and which needs to be investigated in future studies. We have modified the text in the revised manuscript that can be seen in the response to Point 71.

*74.     Page 5, line 169: you filled the gas-exchange tank with SSW. I think you should introduce processes that lead to the establishment of a SML from SSW components, since you saw a relationship for SA in the two compartments. In particular, you should describe processes leading to higher SA and CDOM in the SML. However, you also state that SML and SSW are compositionally distinct. I think this may be contradictory here unless you specify/introduce enrichment and modification processes leading to different organic composition of the two compartments.*

Reply: While we observe distinct compositional differences between the SML and the SSW Cunliffe et al. (2009 and 2013) have demonstrated that the SML can quickly reestablish itself. There is also no practical procedure for collecting large volumes of sample seawater that preserves the integrity of the SML. It is perhaps then even more remarkable that we observe strong and significant relationships with our R660 estimates and SA in situ. To clarify this point we have added a statement in the revised manuscript, which now reads *"We used SSW in the tank experiments for two reasons. First, there is no practical procedure for collecting a large volume sample of surface seawater that preserves the integrity of the SML. Second, we have shown (i) that following its disturbance by vigorous mixing in a laboratory tank the SML becomes re-established on a time scale of seconds with respect to surfactants and other SML components (Cunliffe et al., 2013); (ii) that a new SML is similarly established when sub-surface coastal waters are pumped into large mesocosm tanks (Cunliffe et al., 2009)."* (Lines 139-143).

*75.     Line 175: why don't you try specific CDOM a(λ) to check for correlation to R660 instead of a mean value over a large wavelength range? Line 193: which correlation? Between R660 in the tank and SA of SML in situ?*

Reply: We had completed this analysis but found no apparent relationships. This is now stated in the revised manuscript on lines 250-253.

*76.     Page 6 line 195: I think this second argument does not add relevant information to your data and there is a lack of arguments to explain your observed correlation (that I*

*suppose being between R660 and SA in field SML). You should instead refer to studies linking the composition and/or temporal dynamics of SML with SSW and processes leading to the enrichment of DOM in he SML. You could rather emphasize that you saw a correlation in SA between SML and SSW, which has been shown in many studies, and make references to those.*

Reply: We have now modified this text in the revised manuscript to fully explain our hypothesis, which now reads *"Given that our methodological approach was specifically designed to constrain the effect of surfactants in the SML on $k_w$ and that this minimized the effects of other potential $k_w$ controls, our observations of distinct changes in the quantity and composition of OM in the SML and SSW prompt us to hypothesize that the observed spatio-temporal variation in $R_{660}$ and its relationship with SA (Fig. 3) is a consequence of compositional differences in the surfactant fraction of the SML DOM pool. The principal driver of this hypothesis is the data scatter inherent in the relationship between $R_{660}$ and SA. While we have not been able to unequivocally relate any control of $k_w$ to CDOM absorbance characteristics, and by inference CDOM composition, we nevertheless hypothesise that a rigorous characterisation of the chemical composition of the surfactant pool will yield important insights into surfactant sources and biogeochemical processing that, when analysed in the context of physical forcing such as variable wind regime and hydrography (e.g. Chen et al., 2013; Frew et al., 2006; Gasparovic et al., 2007; Lechtenfeld et al., 2013), will inform a better understanding of the spatial and temporal variability in $k_w$."* (Lines 268-278).

*77.    Line 203: please give examples with the aid of the literature of what kind of organic components, both from biological and anthropogenic sources, can be part of the surfactant fraction of the SML.*

Reply: We have now included the types of organic components and their possible sources in the Introduction section of the revised manuscript (Lines 40-52; see response to Point 31).